# Renormalised spectral flows

Jens Braun[1,2,3], Yong-rui Chen[4], Wei-jie Fu[4], Andreas Geißel[1], Jan Horak[5],
Chuang Huang[4], Friederike Ihssen[5], Jan M. Pawlowski[2,5], Manuel Reichert[6],
Fabian Rennecke[7,8], Yang-yang Tan[4], Sebastian Töpfel[1,3],
Jonas Wessely[5] and Nicolas Wink[1]

1 Institut für Kernphysik, Technische Universität Darmstadt, D-64289 Darmstadt, Germany
2 ExtreMe Matter Institute EMMI, GSI, Planckstraße 1, D-64291 Darmstadt, Germany
3 Helmholtz Research Academy Hesse for FAIR, Campus Darmstadt,
D-64289 Darmstadt, Germany
4 School of Physics, Dalian University of Technology, Dalian, 116024, P.R. China
5 Institut für Theoretische Physik, Universität Heidelberg,
Philosophenweg 16, 69120 Heidelberg, Germany
6 Department of Physics and Astronomy, University of Sussex, Brighton, BN1 9QH, U.K.
7 Institut für Theoretische Physik, Justus-Liebig-Universität Gießen, 35392 Gießen, Germany
8 Helmholtz Research Academy Hesse for FAIR, Campus Gießen, 35392 Gießen, Germany

## Abstract

We derive renormalised finite functional flow equations for quantum field theories in real
and imaginary time that incorporate scale transformations of the renormalisation condi-
tions, hence implementing a flowing renormalisation. The flows are manifestly finite in
general non-perturbative truncation schemes also for regularisation schemes that do not
implement an infrared suppression of the loops in the flow. Specifically, this formulation
includes finite functional flows for the effective action with a spectral Callan-Symanzik
cutoff, and therefore gives access to Lorentz invariant spectral flows. The functional
setup is fully non-perturbative and allows for the spectral treatment of general theories.
In particular, this includes theories that do not admit a perturbative renormalisation
such as asymptotically safe theories. Finally, the application of the Lorentz invariant
spectral functional renormalisation group is briefly discussed for theories ranging from
real scalar and Yukawa theories to gauge theories and quantum gravity.

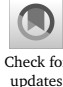
# 1 Introduction

Many interesting non-perturbative phenomena in physics are inherently timelike, ranging from scattering processes, the formation and spectrum of bound states to the time evolution of quantum systems close and far from equilibrium. The qualitative and, even more, quantitative description requires non-perturbative numerical approaches to real-time quantum field theories. In the past years, the functional renormalisation group (fRG) approach, [1] and [2,3], has proven to be a very valuable framework in this context. For a recent general fRG review see [4], for real-time applications of this approach in a broad variety of research fields, see, e.g., [5–27].

In the present work we set up a *finite renormalised* fRG approach. The renormalisation is transported along with the flow and hence is called *flowing renormalisation*. One of its

advantages is its manifest finiteness, also for regulators or regularisation schemes that do not directly implement a UV decay in the loops of the flow equation. This allows for its application to general non-perturbative truncation schemes.

In particular, we use this novel fRG setup to derive a spectral fRG approach in real time, the *spectral fRG*. Our approach is based on the spectral representation of correlation functions, and is manifestly finite as well as Lorentz (or Galilei/Schrödinger) invariant and gauge consistent. It builds on the novel functional spectral approach setup [28,29] which has already been used in [24].

Here, we specifically concentrate on its derivation from finite fRG flows with spatial momentum regulators in the limit where these regulators turn into masslike Callan-Symanzik (CS) regulators. Importantly, this allows for a derivation of the spectral fRG for the effective action in a well-defined spectral way, while full Lorentz invariance is smoothly achieved in the CS limit. In particular, no regularisation of non-perturbative diagrams is implied, but all diagrams discussed are manifestly finite. The valuable benefit of gauge consistency is guaranteed as in the CS limit no momentum cutoff is involved. Finally, its real-time nature allows for an on-shell renormalisation scheme which also facilitates computations. In summary, the present work lays the methodological ground for spectral fRG studies of real-time quantum field theories including QCD and quantum gravity, which is illustrated within example theories ranging from real scalar and Yukawa theories to gauge theories and quantum gravity.

In section 2 we discuss standard fRG flows and the preservation of Lorentz invariance, causality and finiteness within given regularisation schemes. In section 3 we derive the finite fRG flow with *flowing renormalisation* including finite Callan-Symanzik equations. In section 4 the novel setup is used to derive the Lorentz invariant spectral fRG. Finally, in section 5 we show how to use the spectral fRG in various theories. Its practical application in the lowest order of the derivative expansion including flowing renormalisation is discussed in appendix A. We close with a short discussion in section 6.

## 2 Functional renormalisation group

In this section we introduce the fRG equation for the scale-dependent (one particle irreducible) effective action. In section 2.1, we briefly review the standard fRG approach with the inclusion of general renormalisation group transformations into the functional flow. In section 2.2 we discuss the breaking or preservation of space-time symmetries as well as finiteness of the flows for given classes of regulators.

This flow is then used to derive the manifestly finite CS equation (39), on which the spectral fRG is built. In section 3.2, we construct flows with spatial momentum regulators that also feature an ultraviolet cutoff scale. There are then used to define the well-defined Callan-Symanzik limit of these flows.

### 2.1 fRG flow for the effective action

We consider the fRG equation for the one-particle irreducible (1PI) effective action of a generic quantum field theory with a (super) field $\Phi = (\Phi_1, ... \Phi_n)$ comprising all fields. For example, in an $O(N)$ theory we have $\Phi = (\phi_1, ..., \phi_N)$. In QCD, the super field is given by, e.g., $\Phi = (A_\mu, c, \bar{c}, q, \bar{q})$ with gluons $A_\mu$, ghosts $c, \bar{c}$, and quarks $q, \bar{q}$. In gravity, we have, e.g., $\Phi = (h_{\mu\nu}, c_\mu, \bar{c}_\mu)$ with the graviton $h_{\mu\nu}$ and ghosts $c_\mu, \bar{c}_\mu$.

The fRG equation for the scale-dependent effective action $\Gamma_k[\Phi]$ is an exact equation, which expresses the dependence of the (1PI) effective action on an infrared cutoff $k$. It has a simple,

closed one-loop form [1],

$$\partial_t \Gamma_k[\Phi] = \frac{1}{2} \text{Tr}\, G_k[\Phi]\, \partial_t R_k, \qquad t = \log\left(\frac{k}{k_{\text{ref}}}\right), \tag{1}$$

with the (negative) RG time $t$, where the logarithm of $k$ is measured relative to some reference scale $k_{\text{ref}}$. The regulator $R_k$ is a matrix in field space, typically diagonal for bosonic fields and symplectic for fermions, with entries $R_k^{\Phi_i}$. The regulator is the key ingredient in the suppression of infrared physics via the cutoff $k$ and is discussed in the following. The reference scale is usually either chosen to be the initial cutoff scale $k_{\text{init}}$ in the ultraviolet or a physics scale in the theory such as the mass gap or $\Lambda_{\text{QCD}}$ in QCD. The relation of this RG equation to the flow equation of the Wilson effective action, the Polchinski equation [30], has been detailed in [2,3,31].

The flow equation (1) depends on the full regulator-dependent propagator:

$$G_k[\Phi] = \frac{1}{\Gamma_k^{(2)}[\Phi] + R_k}, \quad \text{with} \quad \Gamma_k^{(n)}[\Phi] = \frac{\delta^n \Gamma_k[\Phi]}{\delta\Phi\cdots\delta\Phi}. \tag{2}$$

The functional flow equation (1) entails the scale dependence of the full effective action, while that of the 1PI correlation functions $\Gamma_k^{(n)}$ is obtained by applying the $n$th field derivative to (1). In general, the functional flow of the correlation function $\Gamma_k^{(n)}$ is dependent on $\Gamma_k^{(n+2)}$. The dependence on higher order correlation functions usually has to be truncated in applications.

## 2.2 Infrared regularisation and symmetries

In the following we discuss some properties of the functional flow equation (1) with respect to the choice of regulator. The discussion holds for a generic quantum field theory. We concentrate on the simple example of a real scalar field $\Phi = \phi$ with the classical action

$$S_\phi[\phi] = \int d^d x \left[ \frac{1}{2}\phi\left(-\partial^2 + m_\phi^2\right)\phi + \frac{\lambda_\phi}{4!}\phi^4 \right]. \tag{3}$$

The regulator is introduced into the theory as an infrared modification of the classical dispersion,

$$S_\phi[\phi] \to S_\phi[\phi] + \frac{1}{2}\int_p \phi(p) R_k^\phi(p) \phi(-p), \tag{4}$$

with $\int_p = \int d^d p/(2\pi)^d$. The addition of the regulator term (4) alters the dispersion relation,

$$p^2 + m_\phi^2 \to p^2 + m_\phi^2 + R_k^\phi(p). \tag{5}$$

The derivative of the resulting $k$-dependent generating functional of the theory with respect to $k$ leads to the flow (1), derived in [1].

The regulator can be parametrised with

$$R_k^\phi(p) = Z_\phi k^2 r^\phi(x), \qquad x = \frac{p^2}{k^2}, \qquad \text{or} \qquad x = \frac{\vec{p}^{\,2}}{k^2}, \tag{6}$$

where $Z_\phi$ is the cutoff dependent wave function renormalisation of the field at hand. The *shape function* $r(x)$ depends on either full or spatial momenta squared, $p^2$ or $\vec{p}^{\,2}$, measured in the cutoff scale $k^2$. It implements both, the vanishing momentum limit associated with an infrared (IR) mass as well as the ultraviolet (UV) decay,

$$\lim_{x\to 0} r(x) = 1, \qquad \lim_{x\to\infty} x^{d/2} r(x) \to 0, \tag{7}$$

see e.g. [32] for a respective discussion. The first property implements IR regularisation through an, in general momentum-dependent, mass term that effectively suppresses quantum fluctuations of field modes with momenta $p^2 \lesssim k^2$. The second property leads to a suppression of modes with $p^2 \gtrsim k^2$ in the momentum-loop integrals, rendering the flow (1) and all its field derivatives, which yield the flow of (1PI) correlation functions, UV finite. A specific example for a smooth shape function is

$$r_{\exp}^{\phi}(x) = e^{-x}.\tag{8}$$

In addition to the conditions in (7), which guarantee the finiteness of fRG flows, we might want to impose additional, physically motivated conditions onto the regulators. For relativistic theories it is desirable that the regulators do not spoil Lorentz/Poincaré invariance. Furthermore, for studies of real-time properties, i.e. in Minkowski space, causality should also not be violated. The latter is directly related to the existence of a spectral representation for the propagator of $\phi$.

To maintain Lorentz invariance, the regulator should be a function of the four-momentum squared, $R_k^{\phi}(p^2)$. However, as discussed, e.g., in [7], such regulators might spoil causality through unphysical poles in the complex frequency plane. Typically, such regulators either do not admit a spectral representation or generate fictitious mass poles that only disappear in the vanishing cutoff limit, for a discussion of the latter see [7, 10, 14]. As an example, consider a classical Euclidean propagator

$$G_{\phi,k}(p) = \frac{1}{p^2 + m_{\phi}^2 + R_k^{\phi}(p)},\tag{9}$$

with a regulator shape function, c.f. (6),

$$r_{\mathrm{rat}}^{\phi} = \sum_{n=n_{\min}}^{n_{\max}} c_n \left(\frac{k^2}{k^2 + p^2}\right)^n.\tag{10}$$

Already for such a simple propagator, the existence of a spectral representations of the regularised propagator is highly dependent on the coefficients $c_n$, and in general not the case, see [7, 33] for more details. For general propagators, regulators of the type (10) typically generate at least $n_{\max}$ poles in the propagator, whose positions in the complex plane usually spoil the spectral representation. Another choice would be a variation of the exponential regulator (8), see [10, 14] for more details. Regulators of this type lead to series of poles in the propagator as well as an essential singularity at infinity.

A further common choice are regulators that only depend on the spatial momenta, $R_k^{\phi}(\vec{p}^{\,2})$. Clearly, these regulators do not lead to additional poles in the complex frequency plane, but merely modify the dispersion of the fields. Thus, they admit a spectral representations at the cost of violating Lorentz invariance. If the system is in a medium, explicit Lorentz symmetry breaking might seem innocuous, as it is broken anyway. While this has been confirmed in specific examples [14, 34], it is *a priori* unclear in general. Especially when considering limiting cases of a phase diagram such as $T \to 0$, the question becomes much more intricate than the comparisons in the aforementioned works. Hence, effectively we either violate (or at least complicate) causality, or we violate Lorentz invariance. All known examples of regulators rely on the regularisation conditions in (7). However, by relaxing at least one of these conditions, there is a natural choice for a regulator which preserves both causality and Lorentz invariance,

$$R_{k,\mathrm{CS}}^{\phi} = Z_{\phi}\,k^2, \qquad r_{\mathrm{CS}}(x) = 1,\tag{11}$$

which we refer to as the CS regulator. It implements IR regularisation through an explicit mass $\Delta m_{\phi}^2 = Z_{\phi}\,k^2$. In this case the flow equation (1) has been derived in [35]. To our knowledge,

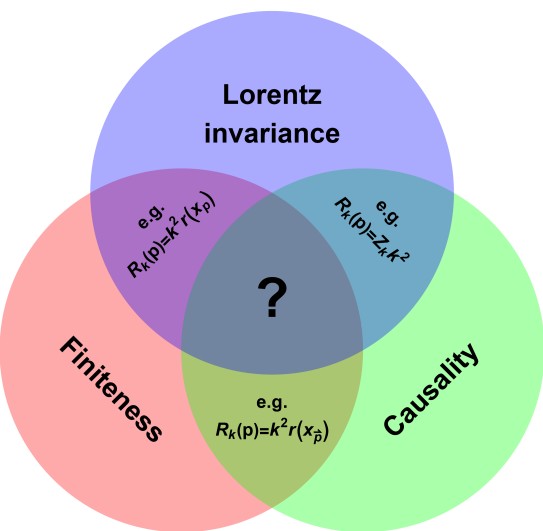

Figure 1: Sketch of the competing requirements for regulators: finiteness of the flow, Lorentz invariance and causality of regulators. Examples for regulators with two of the properties are given. A fully systematic construction of regulators with all three properties in the flow is lacking to date.

it is indeed the first occurrence of such a closed (and one loop) exact functional equation for the effective action. The functional CS flow has been picked up and discussed later in [36–41] as a special choice of the general flow equation (1) .

The insertion of the CS regulator in (1) leads us to the (inhomogeneous) functional CS equation. However, it violates the second condition (7). The CS regulator only lowers the UV degree of divergence by two, for example, quadratically divergent diagrams such as the tadpole diagram in the two point function of the $\phi^4$ theory in $d = 4$ leads to logarithmically divergent tadpole diagrams in the CS equation. In short, at each $k$ in the flow, all loop momenta contribute. To render the flow finite, an additional UV regularisation is required in general.

The different properties of the regularisation are summarised in fig. 1. Restricting the discussion to vacuum for simplicity, the three different property of interest are

1. **Lorentz invariance:** The regulator is a function of $p^2$ and respects Lorentz symmetry.

2. **Causality:** The *regularised* propagator admits a spectral representation, c.f. (42). Expressed in Euclidean momenta, the right half-plane for analytically continued momenta is holomorphic.

3. **Finiteness:** All diagrams arising from (1) and its functional derivatives are finite.

In the overlap regions of fig. 1 we provide examples for regulators with the respective two properties.

No example is given in the overlap regime in the middle with all three properties: at present, no regulator keeping all three properties in fig. 1 simultaneously is known: indeed, the structure of the full propagator,

$$G_k(p) = \frac{1}{\Gamma_k^{(2)}(p) + R_k(p)}, \tag{12}$$

which is the inverse of the regulator and the (yet to be determined) two point function $\Gamma_k^{(2)}$ entails that a systematic construction of such a regulator for all cutoff scales $k$ necessarily

requires the use of the complex structure of $\Gamma_k^{(2)}$ in the regulator. This leaves us with a combination of requirements: the existence of the spectral representation of the propagator (12) with the regulator limits (7) for an unknown two-point function $\Gamma_k^{(2)}$. This combination is rather obstructive, and if a systematic construction is possible at all, it evidently requires using constraints on the complex structure of the two point function at hand.

As an illustration, let us assume we have a Lorentz invariant regulator with the properties (7). We observe that the regulator needs to be a decaying function as $p \to \infty$, by assumption, while $\Gamma^{(2)}(p)$ needs to be a growing function for $p \to \infty$, being the inverse of the propagator. This implies finite Lorentz invariant flow equations (properties 1 & 3). Now we show that then causality (property 2) is at stake:

A simple consideration of the Cauchy–Riemann equations suggest different signs of their imaginary parts in the top-right quadrant of $p \in \mathbb{C}$. However, the regulator needs to have a positive real part, at least for small Euclidean momenta, to provide the IR regularisation. In a partially simplified picture, this leads to lines in the complex plane where the real part of the regulator is zero. Similarly, the real part of the two-point function has lines with vanishing real part, related to the dispersion relation. The different limiting cases detailed above make it almost impossible to avoid zeros in the top right quadrant of the complex momentum plane, and consequently lead to a violation of causality in the regularised propagator. Partly, this reasoning can also be found in [25]. The argument presented here is a very pictorial, simplified version. While it is easy to construct explicit counterexamples, so far even in tailor made applications, such as spectral functions of a simple scalar theory, no regulator has been provided that escapes this problem, leave aside a generic systematic construction scheme. A full discussion of this issues is postponed to future work.

This leaves us with the situation that we may consider regulators in the three overlap regions, put differently, regulators, that lack one of the properties 1-3. In this context we emphasise a peculiarity of the overlap regime without finiteness including the CS regulator: the structural similarity of the Wetterich equation (1) with regulators obeying (7) and the flow with the CS regulator (11) is misleading. While the former equation implements a Wilson-type momentum-shell integration in a fixed underlying quantum field theory, the CS flow constitutes a flow in the space of theories. To be specific, the need for additional UV regularisation at different cutoff scales $k$ implies that we have different theories which necessarily require a different renormalisation. Hence, the flow must be re-renormalised; only specifying the initial effective action $\Gamma_{k_{\mathrm{init}}}$ does not lead to a finite renormalised solution of the flow equation. This renormalisation is typically done with an RG transformation, leading to finite renormalised loops as well as the $\beta$-function and anomalous dimension terms in (26). As we will show in the next section, this can be achieved instead by the introduction of explicit counter terms to the flow, supplemented with renormalisation conditions which are fixed at a, in general $k$-dependent, renormalisation scale $\mu$.

# 3 Functional flows with flowing renormalisation

In this section we discuss the finiteness of infrared flows and the finiteness of the ultraviolet limit of the effective action. Both properties are related to the UV renormalisation that is implicitly or explicitly implemented in the flow equation. This leads us to the concept of flowing renormalisation.

In section 3.1 we discuss infrared RGs with general rescalings during the RG flows and introduce the concept of RG-consistency. This setup allows us to define flows that have a finite UV limit, hence elevating the standard *bare* diverging UV effective action to a *renormalised* finite UV action. In section 3.2 we then derive a key result of the present work, the general

flow equation (38) with a flowing Bogoliubov-Parasiuk-Hepp-Zimmermann (BPHZ)-type UV renormalisation. This renormalised fRG flow is not based on full RG rescalings which imply a multiplicative renormalisation that typically cannot be used in non-perturbative truncation schemes. This is achieved by augmenting the infrared RG steps with explicit ultraviolet ones that are also formulated in terms of a standard functional RG. In section 3.3 we discuss the properties of the manifestly finite CS equation (39) obtained from (38) for the CS regulator including the implementation of general renormalisation conditions, which is the basis for the spectral fRG discussed in section 4. In section 3.4 we summarise the results and findings of this Section and emphasise some important aspects.

## 3.1 RG-consistency and UV scaling

For standard infrared regulators with shape functions $r(x)$ that obey (7), the flow equation is manifestly finite as loop momenta are effectively restricted to $p^2 \lesssim k^2$. Then, choosing a specific $r(x)$ amounts to specifying a UV regularisation scheme for fRG flows. The effective action $\Gamma_k$ of a general theory is then obtained by integrating (1) from some initial cutoff scale $k_{\text{init}}$ to $k \le k_{\text{init}}$,

$$\Gamma_k[\Phi] = \Gamma_{k_{\text{init}}}[\Phi] + \int_{k_{\text{init}}}^{k} \frac{dk'}{k'} \partial_t \Gamma_{k'}[\Phi]. \tag{13}$$

The renormalisation conditions are implicitly fixed through the initial effective action $\Gamma_{k_{\text{init}}}$. The theory at a given cutoff scale $k$ should not depend on the initial cutoff scale $k_{\text{init}}$, which is called *RG-consistency*, see [32, 42, 43],

$$k_{\text{init}} \frac{d\Gamma_k}{dk_{\text{init}}} = 0. \tag{14}$$

Since the initial effective action implicitly fixes the renormalisation conditions, RG-consistency entails renormalisation group invariance, and specifically the independence of the theory on these conditions. Inserting (13) into (14), we arrive at

$$\partial_{t_{\text{init}}} \Gamma_{k_{\text{init}}}[\Phi] = \frac{1}{2} \text{Tr}\, G_k[\Phi] \, \partial_t R_k \Big|_{k_{\text{init}}}. \tag{15}$$

Equation (15) entails that the $k_{\text{init}}$-dependence of the effective action at the initial (large) cutoff scale is given by the flow equation. Hence, the running of the UV relevant parameters can be read off from the IR flow equation for asymptotically large cutoff scales, where the flow of a given coupling is proportional to

$$\lim_{k \to \infty} \partial_t \lambda_i \propto k^{d_{\lambda_i}}. \tag{16}$$

The right hand side includes the full $k$-scaling: the combination of the scaling of the loop integrals and the vertices. Then, UV relevant and marginal couplings $\lambda_i$ have scaling dimensions $d_{\lambda_i} \ge 0$, while UV irrelevant couplings have scaling dimensions $d_{\lambda_i} < 0$. Consequently, for asymptotically large infrared cutoff scales, the effective action approaches the *bare* UV effective action: Only the UV relevant operators survive and diverge with $k \to \infty$ according to their scaling dimension with $k^{d_{\lambda_i}}$ for $d_{\lambda_i} > 0$ and logarithmically with $t$ for $d_{\lambda_i} = 0$.

As a first step towards the desired finite flow equations, also for regulators such as the CS one, we discuss how the UV scaling (16) can be absorbed within a general RG rescaling. Then, the UV limit of the effective action is the finite renormalised UV action and not the diverging bare action. For more details we refer the reader to [32] and in particular [44].

The underlying RG invariance of the theory at $k = 0$ implies that the full effective action $\Gamma = \Gamma_{k=0}$ obeys the homogeneous renormalisation group equation

$$\mu \frac{d\Gamma[\Phi]}{d\mu} = \left( \mu \partial_\mu + \beta^{(\mu)}_{\lambda_i} \partial_{\lambda_i} + \int_x \gamma^{(\mu)}_{\Phi_j} \Phi_j \frac{\delta}{\delta \Phi_j} \right) \Gamma[\Phi] = 0 \,, \tag{17}$$

where $\int_x = \int d^d x$ refers to the integral over spacetime. For a detailed discussion see Chapter IV in [32]. The $\beta$-functions $\beta^{(\mu)}$ and anomalous dimensions $\gamma^{(\mu)}_{\Phi_j}$ of the theory at hand are defined as

$$\gamma^{(\mu)}_\Phi \Phi = \mu \frac{d\Phi}{d\mu} \,, \qquad \beta^{(\mu)}_\lambda = \mu \frac{d\lambda}{d\mu} \,. \tag{18}$$

The coupling vector $\lambda = (\lambda_1, ..., \lambda_m)$ contains all relevant parameters of the theory, including the mass parameters. Note that equation (17) entails the invariance of the underlying quantum field theory under self-similarity transformations of the theory.

It has been shown in Chapter VIII B of [32], that the RG invariance of the theory is maintained in the scale-dependent theory in the presence of the regulator term for regulators of the form of (6). Such regulators are called *RG-adapted* as they satisfy the RG equation

$$\left( \mu \partial_\mu + 2\gamma^{(\mu)}_\Phi \right) R^\Phi_k = 0 \,, \tag{19}$$

and scale as an inverse two-point function, see (8.14) in [32]. The respective full RG equation reads

$$\left( \mu \partial_\mu + \beta^{(\mu)}_{\lambda_i} \partial_{\lambda_i} + \int \gamma^{(\mu)}_{\Phi_j} \Phi_j \frac{\delta}{\delta \Phi_j} \right) \Gamma_k[\Phi] = 0 \,, \tag{20}$$

and has the same form as the RG equation at $k = 0$, (17). From this, we obtain the general flow equation that comprises the change of a cutoff scale, here $k$, as well as an accompanying general RG transformation. We remark that general reparametrisations (self-similarity transformations) can also involve non-linear field transformations, $\Phi_i \to \phi_i[\Phi]$, which might facilitate the discussion of the renormalisation in specific cases. This has been considered in Chapter VII A and B of [32] and in [45] for the Wilsonian effective action. More recently, these general field redefinitions have been used for setting up *essential* fRG flows in [46, 47].

Using (1), an additional total $k$-derivative of (20) yields the flow equation with reparametrisation at each flow step, see (4.25) in Chapter IV of [32],

$$\left( s \partial_s + \beta^{(s)}_{\lambda_i} \partial_{\lambda_i} + \int_x \gamma^{(s)}_{\Phi_j} \Phi_j \frac{\delta}{\delta \Phi_j} \right) \Gamma_k[\Phi] = \frac{1}{2} \mathrm{Tr}\, G_k[\Phi] \left( \partial_s + 2\gamma^{(s)}_\Phi \right) R^\Phi_k \,, \tag{21}$$

where we consider $k(s)$ and $\mu(s)$ with

$$s \partial_s = \mu \partial_\mu + \partial_t \,. \tag{22}$$

The $\beta$-functions $\beta^{(s)}$ and anomalous dimensions $\gamma^{(s)}$ then encode the full $s$-scaling of a combined cutoff ($k$-) and RG ($\mu$-) flow, including a reparametrisation of the theory,

$$\gamma^{(s)}_\Phi \Phi = s \frac{d\Phi}{ds} \,, \qquad \beta^{(s)}_\lambda = s \frac{d\lambda}{ds} \,. \tag{23}$$

Hence, the loop term on the right hand side of (21) is proportional to the full $s$-scaling of the cutoff term, consisting of the infrared cutoff scaling with $k$, the renormalisation group

scaling with $\mu$ and a potential scaling of an UV cutoff scale $\Lambda$. This $s$-scaling reduces to (18) for $s\partial_s\mu = \mu$ and $s\partial_s k = 0$, and to the standard fRG anomalous dimension and $\beta$-functions for $s\partial_s k = k$ and $s\partial_s\mu = 0$. Finally, the linear combination (22) of $k$ and $\mu$ scalings leads to $\gamma_\Phi^{(s)} = \gamma_\Phi^{(\mu)} + \gamma_\Phi$ with

$$\gamma_\Phi \equiv \gamma_\Phi^{(k)} = -\frac{1}{2}\partial_t \log Z_\Phi\,. \tag{24}$$

For RG-adapted regulators with (19) the renormalisation group scaling drops out of the right hand side. For example, for the linear combination of the two scalings with $k$ and $\mu$ we arrive at

$$\begin{aligned}
\left(\partial_s + 2\gamma_\Phi^{(s)}\right)R_k^\Phi &= \left(\partial_t + 2\gamma_\Phi\right)R_k^\Phi + \left(\mu\partial_\mu + 2\gamma_\Phi^{(\mu)}\right)R_k^\Phi \\
&= \left(\partial_t + 2\gamma_\Phi\right)R_k^\Phi\,. \tag{25}
\end{aligned}$$

With (25) the flow (21) reduces to

$$\left(s\partial_s + \beta_{\lambda_i}^{(s)}\partial_{\lambda_i} + \int_x \gamma_{\Phi_j}^{(s)}\Phi_j\frac{\delta}{\delta\Phi_j}\right)\Gamma_k[\Phi] = \frac{1}{2}\mathrm{Tr}\,G_k[\Phi]\left(\partial_t + 2\gamma_\Phi\right)R_k^\Phi\,. \tag{26}$$

The occurrence of $\gamma_\Phi$ in (26) stems from the linear field-reparametrisation $\Phi \to Z_\Phi^{1/2}\Phi$. For non-linear reparametrisations, the anomalous dimensions are field-dependent, $\gamma_\Phi^{(s)}\Phi \to \gamma^{(s)}[\Phi]\Phi$, and the anomalous dimension in the flow term on the right-hand side of (26) has to be substituted by

$$\gamma_\Phi^{(s)}R_k^\Phi \to \frac{\delta(\gamma^{(s)}[\Phi]\Phi)}{\delta\Phi}R_k^\Phi\,, \qquad \gamma^{(s)}[\Phi]\Phi = s\frac{d\Phi}{ds}\,, \tag{27}$$

with matrix valued anomalous dimensions (in field space). The respective derivations can be found in Chapter VIII A of [32]. The general flow is given by (8.8) in this Chapter. For field-independent $\gamma^{(s)}$, (27) boils down to the standard expressions. In any case, due to reparametrisation invariance, such reparametrisations are optional and might be used to simplify certain computations, e.g., in the context of critical physics, where anomalous dimensions are of central interest.

Equation (26) including the field-dependent generalisation (27) is the general fRG setup for the effective action. For regulators with the second property (UV decay), general reparametrisations encoded in the anomalous dimensions and $\beta$-functions may facilitate the computations or implement functional optimisation schemes. In particular, we can absorb the UV scaling (16) of the UV-relevant couplings with $d_{\lambda_i}$ into the anomalous dimensions and $\beta$-functions, leading to a finite renormalised UV effective action. This is simply a convenience for infrared flows with finite flow equations, but is a necessity in the absence of ultraviolet finite loops, as is the case for the CS regulator, (11). Then, the rescalings implement the required UV renormalisation via multiplicative renormalisation. While this is a formally correct procedure, the implementation of multiplicative renormalisation within non-perturbative truncation schemes is intricate. This intricacy is present for all diagrammatic methods such as DSEs or 2PI methods, a detailed discussion is provided in [48].

As the central result of this paper, we will show in the next section that the additional $\mu$-flow can be absorbed into a well-defined flow of a non-perturbative counter term action for the $k$-flow in a manifestly finite way. The flow of the counter term action serves a two-fold purpose: Firstly, it allows to change consistently the renormalisation conditions with the $k$-flow for general IR flows. We call this flowing renormalisation. Secondly, it also leads to manifestly finite flows for the CS regulator with a flowing counter term action for general non-perturbative truncation schemes. The number of parameters in these counter term matches that of relevant parameters in the theory.

## 3.2 Functional RG with flowing renormalisation

We now use the general flow equation with an infrared regulator and an ultraviolet one for deriving a flow equations which also incorporates an explicit UV renormalisation in a manifestly finite approach in terms of a generalised BPHZ scheme with the subtraction of a flowing counter term action. In contradistinction to multiplicative schemes this leads to finite loop diagrams by subtraction. Such a construction has the benefit of a simple and robust numerical implementation.

This general setup also allows us to monitor and change the renormalisation conditions within the infrared flow. This generalises the standard fRG setup, in which the (UV) renormalisation and the respective renormalisation conditions are implicit in the choice of the finite initial action, see the discussion around (13).

The access to the UV behaviour of the theory is obtained by introducing a regulator $R_{k,\Lambda}(p)$, where an UV cutoff $\Lambda = \Lambda(k)$ enters as a free parameter/function. The regulator $R_{k,\Lambda}^{\Phi}$ is chosen such that it effectively restricts loop momenta to $p^2 \lesssim \Lambda(k)^2$ in the loops of the flow equation, see the examples (30c) and (30d) below. We may also use the regulator for a full UV regularisation of the theory and not only the flow equation, see e.g. (30e) below.

Changing the UV scale $\Lambda = \Lambda(k)$ alongside with the infrared flow allows us to introduce a *flowing* (UV) renormalisation in the latter. For these regulators the flow (1) can be written as

$$\left(\partial_t\big|_{\Lambda} + \mathcal{D}_k \, \partial_{t_{\Lambda}}\right)\Gamma_{k,\Lambda} = \frac{1}{2}\mathrm{Tr}\, G_{k,\Lambda}^{\Phi}\left(\partial_t\big|_{\Lambda}R_{k,\Lambda}^{\Phi} + \mathcal{D}_k\, \partial_{t_{\Lambda}}R_{k,\Lambda}^{\Phi}\right), \tag{28}$$

where $t_{\Lambda} = \log(\Lambda/k_{\mathrm{ref}})$, with a reference scale $k_{\mathrm{ref}}$. The factor $\mathcal{D}_k$ is a relative measure of RG steps in the $k-$ and the $\Lambda$-direction,

$$\mathcal{D}_k = \partial_t \log \Lambda(k). \tag{29}$$

The flow (28) is a finite functional flow which allows us to successively integrate out momentum shells. For $\partial_{t_{\Lambda}}R_{k,\Lambda}^{\phi} = 0$ we arrive at the standard (infrared) flow in (1). This naive limit can only be taken for infrared momentum cutoffs that decay sufficiently fast in the ultraviolet. Most importantly, we can identify the terms $\propto \mathcal{D}_k$ in (28) as UV-cutoff flows that can be used for a flowing renormalisation scheme.

This derivation holds true for general infrared regulators. In the following we use as an important example regulators $R_{k,\Lambda}^{\phi}$, that converge towards the CS regulator with the shape function (11) for $\Lambda \to \infty$. In this case the flowing renormalisation can now be used to derive the finite fRG flow (28) for the CS regulator. For this derivation it is convenient to consider regulators $R_{k,\Lambda}^{\phi}$ with

$$R_{k,\Lambda}^{\phi}(p) = Z_{\phi}\, k^2\, r(x_{\Lambda}), \qquad x_{\Lambda} = \frac{\vec{p}^{\,2}}{\Lambda^2}, \tag{30a}$$

where we have considered a spatial momentum regulator in order to retain causality in a simple manner, as discussed in the previous section. Again, we emphasise that this choice is only taken for the sake of the spectral flows discussed later, it is not a necessary one. For $\Lambda \to \infty$ we require

$$\lim_{\Lambda \to \infty} R_{k,\Lambda}^{\phi} = Z_{\phi}\, k^2, \tag{30b}$$

which leaves us with the CS flow as limit of well-defined UV-finite flows. Explicit examples for shape functions are given by

$$r_{\mathrm{exp}}(x_{\Lambda}) = e^{-x_{\Lambda}}. \tag{30c}$$

This regulator leads to an exponential damping of the UV modes in the loop via the regulator in the numerator of the flow. Another regulator of this type is given by

$$r_{\text{cs}}(x_\Lambda) = \theta(1 - x_\Lambda). \tag{30d}$$

Again the loop is rendered finite via the regulator in the numerator of the loop. We emphasise that, (30c) does not imply a UV regularisation of standard diagrams, e.g. in perturbation theory or a system of Dyson-Schwinger equations, but only an UV-regularisation of the loops in the flow equations.

We may augment the IR regulator with a UV regulator, leading to UV and IR finite loops with

$$r_{\text{sharp}}(x_\Lambda) = \frac{1}{\theta(1 - x_\Lambda)} . \tag{30e}$$

This regulator leads to momentum loops, e.g. in perturbation theory or a system of Dyson-Schwinger equations, that do not receive any contribution from spatial loop momenta $\vec{p}^{\,2} > \Lambda^2$. Naturally, this property also holds true for the respective flow equations. All the regulators in (30) and the limit $\Lambda \to \infty$ satisfies the constraint (30b).

To understand the CS limit, we have to explicitly determine the part of the flow that comes from changing the UV cutoff $\Lambda$. For $\Lambda \to \infty$ the second part of the flow,

$$\frac{1}{2}\text{Tr}\, G_{k,\Lambda}^{\phi}\, \mathcal{D}_k\, \partial_{t_\Lambda} R_{k,\Lambda}^{\phi} , \tag{31}$$

takes a simple form: First of all, up to sharply peaked contributions for large momenta, see the examples in (30), the $t_\Lambda$-derivative of the regulator vanishes in the CS limit (30b) with

$$\lim_{\Lambda \to \infty} \partial_{t_\Lambda} R_{k,\Lambda}^{\phi} = 0 . \tag{32}$$

Note that (32) simply entails removing the $\Lambda$-part of the flow in the limit $\Lambda \to \infty$, so it holds true beyond the CS example. Thus, in this limit the contribution of the $\Lambda$-flow, (31), to the full flow, (28), vanishes unless this zero is compensated by a divergence in the $\Lambda$-flow.

On the more technical level we define diagrams with UV *irrelevant* power counting in the flow equation: these are the diagrams $\text{Diag}_i^{(n)}\big(\partial_{t_\Lambda} R_{k,\Lambda}^{\phi}\big)$ in the flow of $n$-point functions $\Gamma_k^{(n)}$ which remains finite if the substitution $\partial_{t_\Lambda} R^{\phi} \to 1$ is done. Here, the superscript $^{(n)}$ indicates a diagram of the flow of $\Gamma_k^{(n)}$, while the subscript $_i$ labels the different diagrams in this flow. We write

$$\lim_{\Lambda \to \infty} \left| \text{Diag}_i^{(n)}\big(\partial_{t_\Lambda} R_{k,\Lambda}^{\phi} \to \Lambda^2\big) \right| < \infty . \tag{33}$$

Diagrams with (33) either contain a sufficiently large number of propagators or sufficiently rapidly decaying vertices to render the integration over loop momenta finite. In the CS limit, the contribution of UV-irrelevant diagrams to the flow vanishes like $\Lambda^{-n}$ with some $n > 0$.

In turn, the power counting marginal and relevant parts of the $\Lambda$-flow (31) will survive in this limit and indeed diverge with powers and logarithms of $\Lambda$. Importantly, these terms are also local *if* the vertices are: they only depend on powers of momenta. A more general analysis of non-perturbative UV renormalisation including also Dyson-Schwinger equations (DSEs) is deferred to [48]. Note also that the $\Lambda$-flow has the same UV power counting as standard diagrams, as the regulator behaves like an inverse propagator for $\Lambda \to \infty$. This can be seen from the example regulators (30), whose $t_\Lambda$-derivative yields

$$
\begin{aligned}
\partial_{t_\Lambda} r_{\text{exp}}(x_\Lambda) &= 2x_\Lambda e^{-x_\Lambda} , \\
\partial_{t_\Lambda} r_{\text{CS}}(x_\Lambda) &= 2x_\Lambda \delta(1 - x_\Lambda) .
\end{aligned}
\tag{34}
$$

Hence, the $\Lambda$-flows for $n$-point functions diverge with the same power of $\Lambda$ as standard loop diagrams, e.g., in perturbation theory. The above analysis leads us to an intricacy of the UV power counting that is elucidated in appendix A at the example of the $\phi^4$-theory in $d = 4$. In short, the standard UV power counting only holds true if the truncation at hand respects the UV power counting of the theory. A prominent important counter example is the derivative expansion in a $\phi^4$-theory. Already its lowest order (0th order derivative expansion or local potential approximation (LPA)) includes a full effective potential $V_{\text{eff}}(\phi^2)$, and hence all order (point-like) interactions $\lambda_n/(n!)\phi^{2n}$ with $n \in \mathbb{N}$. Due to its momentum-independence these couplings persist unchanged at arbitrarily large momentum $p \to \infty$. Accordingly, they seemingly introduce infinitely many fundamental couplings $\lambda_n$, which would render the theory UV-sick. Note that this intricacy also is present for other functional approaches, the Dyson-Schwinger equation for the effective potential in LPA has been discussed in [49], Appendix F, for more details see also [48]. In appendix A it is shown, how the present procedure leads to a well-defined finite result, and the number of relevant parameters matches that in perturbation theory in the UV. A detailed discussion goes beyond the scope of the present paper and is presented in [48], including also non-trivial aspects of momentum dependences of vertices: for non-perturbative approximations with full momentum-dependent vertices the counter terms have a non-trivial but uniquely fixed momentum-dependence. This is similar to the unique non-polynomial field dependence in LPA discussed in appendix A, but in contradistinction to the latter it is no truncation artefact.

Finally, the prefactors of the UV-relevant terms in the $t$ flow may be different from that in the $t_\Lambda$ flow, as the respective scale derivatives of the regulator have a different momentum-dependence if taken for a fixed shape function. This is remedied by using shape functions with

$$r_\Lambda = r(x_\Lambda) + \Delta r_\Lambda(x_\Lambda), \tag{35}$$

and the correction $\Delta r_\Lambda(x_\Lambda)$ is taken such that the relative prefactors of all UV relevant terms of the $t_\Lambda$-flow equals the relative prefactors of the relevant terms in the $t$-flow.

In summary, this leads us to the definition of the counter term action,

$$\partial_t S_{\text{ct}}[\phi] := -\frac{1}{2}\text{Tr}\, G^\phi_{k,\Lambda}\, \mathcal{D}_k\, \partial_{t_\Lambda} R^\phi_{k,\Lambda}, \tag{36}$$

which removes all terms with positive powers $\Lambda^n$ as well as logarithms $\log \Lambda/k_{\text{ref}}$ from (28) and renders the infinite UV cutoff limit finite,

$$\lim_{\Lambda \to \infty} \left|\partial_t \Gamma_{k,\Lambda}[\phi]\right| < \infty. \tag{37}$$

The counter term action (36) depends on a finite set of renormalisation parameters required for the finite limit (37). The size of this set is equivalent to the number of UV relevant and marginal directions. Moreover, in the limit $\Lambda \to \infty$ the counter term action takes a local form for approximations with local vertices that reduce to the classical ones for large momenta.

Finally, we arrive at the novel flow equation with flowing renormalisation

$$\partial_t \Gamma_k[\phi] = \frac{1}{2}\text{Tr}\, G_\phi[\phi]\, \partial_t R^\phi - \partial_t S_{\text{ct}}[\phi], \tag{38}$$

with the flow of the counter term action (36) accounting for the flow of the renormalisation conditions as well as the finiteness for infrared cutoffs such as the CS regulator. This general equation constitutes a main result of our work. It can be augmented with general reparametrisations of the theory, leading to a generalisation of (21): we simply have to subtract $\partial_t S_{\text{ct}}[\phi]$ defined in (36) on the right hand side of (21). Note, that heuristically such a procedure is

suggestive but in general a naive removal of divergent terms does not provide a consistent renormalisation. In the present Section we have proven that (38) is correct. The derivation also offers a systematic practical way to compute the counter term and we shall see in an explicit example that within commonly used truncation schemes this goes beyond using standard counter terms, see in particular appendix A.

### 3.3 Finite CS flows and flowing renormalisation conditions

In the remainder of this work we use this flow for setting up spectral functional flows with the finite CS flows derived from (38). For the CS flow the general equation reduces to

$$\partial_t \Gamma_k[\phi] = \text{Tr} \, G_\phi[\phi] \, k^2 - \partial_t S_{\text{ct}}[\phi], \tag{39}$$

where a CS regulator in a manifestly UV finite setting, such as given with the shape function (30e), is assumed and the finite limit $\Lambda \to \infty$ can be safely taken. As for the general equation (38) the novelty of (39) is not its finiteness per se. Indeed, already the original functional CS equation as derived in [35] can be shown to be finite order by order in perturbation theory. However, (39) is manifestly finite in *general* perturbative and non-perturbative truncation schemes with a manifestly finite effective action. Moreover, the present setup allows for a direct computation of the flow of the counter term action, only dependent on a set of renormalisation parameters which are in one-to-one correspondence to the coefficients of the UV marginal and relevant operators. Finally, the finite CS flow can be applied to perturbatively and non-perturbatively renormalisable theories, and for a first application in quantum gravity we refer to [24].

The general flow (38) and its finite CS limit (39) seemingly imply that we are left with the task of computing the non-trivial scaling factor $\mathcal{D}_k$ as well as the $\Lambda$-trajectory (35) at each RG-step. This would exact a heavy price for the finiteness (37). It is therefore noteworthy that we do not have to compute $\partial_t S_{\text{ct}}[\phi]$ from the flow, as it can completely fixed by the choice of renormalisation conditions. The subtraction $\partial_t S_{\text{ct}}[\phi]$ has to be simply chosen such that the flow of these conditions vanish. This choice is practically implemented by subtracting the $t$-flow of the correlation functions $\Gamma_k^{(n)}(p^2 = \mu^2)$, that is the renormalisation condition from the full $t$-flow. This renders the functional $t$-flow finite (if one also subtracts the zero point function) and guarantees the RG conditions to hold.

We illustrate this within a simple example for the finite CS flow. Again we use a real scalar field theory with the renormalised effective action $\Gamma_{k,\Lambda}$ with a given UV cutoff $\Lambda$. The renormalisation entails that the effective action $\Gamma_{k,\Lambda}$ stays finite in the limit $\Lambda \to \infty$. Moreover, it may satisfy the following on-shell renormalisation conditions at the flowing scale $\mu = \mu(k)$,

$$\lim_{\Lambda \to \infty} \Gamma_{k,\Lambda}^{(2)}[\bar{\phi}](p)\Big|_{p_0^2 = -\mu^2} = -k^2,$$

$$\lim_{\Lambda \to \infty} \partial_{p_0^2} \Gamma_{k,\Lambda}^{(2)}[\bar{\phi}](p)\Big|_{p_0^2 = -\mu^2} = 1,$$

$$\lim_{\Lambda \to \infty} \Gamma_{k,\Lambda}^{(4)}[\bar{\phi}](p)\Big|_{p_0^2 = -\mu^2} = \lambda_\phi, \tag{40}$$

where $p_0$ is the Euclidean frequency and $p^2 = p_0^2 < 0$ is evaluated at a timelike Minkowski momentum with $\vec{p} = 0$ and the Minkowski frequency $\pm\sqrt{-p_0^2}$. Here, $\bar{\phi}$ is a background field, which is typically given by the solution of its (quantum) equation of motion (EoM), $\bar{\phi} = \phi_{\text{EoM}}$.

The first condition is an on-shell mass renormalisation: the effective action in the presence of an IR regulator is defined as a modified Legendre transform excluding the regulator term. Hence, for the physical CS regulator we have to consider the full Euclidean two-

point function with the CS mass term $Z_\phi k^2$, that is $\Gamma_k^{(2)}(p^2) + Z_\phi k^2$. Thus, (40) simply implies $\Gamma_k^{(2)}(-\mu^2) + k^2 = 0$, so the renormalisation scale determines the $k$-dependent pole mass, $\mu = m_k$. By setting $\mu = k$, we can enforce this pole mass to be given by the mass introduced by the CS regulator. Thus, for a given physical mass the RG flow from the initial UV scale $k_{\text{init}}$ is terminated at $k_{\text{fin}} = m_{\text{phys}} = m_{k_{\text{fin}}}$. Put differently, with this RG condition we flow through the space of scalar theories with the physical pole mass $k^2$.

The second condition in (40) fixes the wave function renormalisation at $\mu$, $Z_\phi(-\mu^2) = 1$ on-shell. We remark, that this leads to a spectral function $\rho_{\phi,k}$ that is not normalised to unity if $\phi$ is a physical field (defining an asymptotic state), see section 4 .

The last condition in (40) fixes the quartic interaction vertex. We have not specified the momentum configuration here, but natural choices are the symmetric point and specific momentum channels such as the $s$-channel.

Below, we shall consider more general on-shell as well as off-shell renormalisation conditions adapted to specific theories or classes of theories. We emphasise that every RG-condition serves our purpose, but on-shell RG conditions are in most cases a specifically convenient physical choice, only accessible for real-time formulations.

We also remark that adjusting specific renormalisation conditions in the standard fRG setting is a fine-tuning problem: One has to adjust the initial effective action at the initial cutoff scale $k_{\text{init}}$ such, that the effective action at $k_{\text{fin}}$ satisfies the renormalisation conditions. However, adjusting specific renormalisation conditions is not required in the fRG approach but the same finite tuning task extends to adjusting the physics parameters at the initial scale. Both tasks are solved or at least facilitated in the presence of flowing renormalisation, and (40) exemplifies this general pattern. With (40) both the adjustment of the renormalisation conditions and the adjustment of the physics parameters is done directly.

It is an additional benefit of the present formulation that the usual finite tuning of the physical parameters at $k = 0$ from a set of initial conditions at a large initial cutoff scale $k_{\text{init}}$ can be avoided. From (40) we get

$$\lim_{\Lambda \to \infty} \partial_t \left\{ \Gamma_{k,\Lambda}^{(2)}[\bar{\phi}](p) \Big|_{p_0^2 = -\mu^2} \right\} = -2 k^2 \,,$$

$$\lim_{\Lambda \to \infty} \partial_t \left\{ \partial_{p_0^2} \Gamma_{k,\Lambda}^{(2)}[\bar{\phi}](p) \Big|_{p_0^2 = -\mu^2} \right\} = 0 \,,$$

$$\lim_{\Lambda \to \infty} \partial_t \left\{ \Gamma_{k,\Lambda}^{(4)}[\bar{\phi}](p) \Big|_{p_0^2 = -\mu^2} \right\} = 0 \,, \tag{41}$$

which completely fixes $\partial_t S_{\text{ct}}$ in (39).

We emphasise that the implementation of the above full flowing renormalisation is not required within the formulation. Indeed, in the example of the $\phi^4$ theory in $d = 4$ discussed in detail in appendix A, the only divergence in the flow equation is related to the mass renormalisation: The CS flow lowers the standard UV degree of divergence by two and the field-dependent part of the flow is logarithmically divergent. Thus, the flow of the counter term action $\partial_t S_{\text{ct}}$ only needs to include one term to ensure finiteness. For the explicit LPA example see (A.9) and (A.23) in appendix A. Additional counter terms for further fundamental couplings can still be introduced to enforce the renormalisation conditions. For a scalar theory in LPA, their general form is given by (A.4), and the explicit form of the counter term which renormalises the mass and the coupling of $\phi^4$ theory is shown in (A.32).

We emphasise that using a "minimal" counter term with $k$-independent parameters, i.e. one that only regularisers the divergent contributions, the present approach reduces to the standard infrared flow: the renormalisation group conditions at $k = 0$ are implicitly set at

$k = k_{\text{init}}$ and the physics parameters and RG conditions flow into their final values, which have to be fine-tuned for given physics and RG conditions.

## 3.4 Wrap-up

The derivation of the general fRG flow (38) with flowing renormalisation and its finite CS limit (39) is a key result of the present work. Importantly, the counter-term action $S_{\text{ct}}$ is all that is left from the $\Lambda$ and $\mu$-dependence of the general fRG flow in equation (26). Notably, the conventional finite CS equation also involves the terms proportional to the $\beta$-function on the LHS side of equation (26) (which follow from multiplicative renormalisation), which are missing in equation (39). We emphasise that the latter BPHZ-type renormalisation allows for the implementation of general non-perturbative truncation schemes which are difficult to implement in a setting with multiplicative renormalisation. Put differently, the formal finiteness of the standard CS equation is only of use in truncation schemes such as perturbation theory and does not survive in general non-perturbative truncation schemes.

Equation (38) and (39) can be augmented with $\beta$-function terms. They are present if the flow is amended with an additional standard RG transformation with $\mu(k)$. This is an option in specific cases, as it may facilitate the computations or the convergence of systematic approximation schemes. Still, it is an important result that such an additional RG transformation is not required for finiteness and equation (39) is exact: while the $\beta$-function terms pose no conceptual problem as they can be considered in a *closed* form by auxiliary flows, their computation constitutes in most cases a considerable additional technical challenge. For a detailed discussion of such a setup in a different context, see [50]. There, it is shown how to derive flows for the dependences of vertices or couplings on external parameters such as fundamental couplings, temperature, and chemical potential.

We note that in asymptotically safe theories one may have to consider the non-trivial asymptotically safe momentum scaling of vertices and propagators for the correct propagation of the RG conditions. For example, in asymptotically safe gravity the propagators have an anomalous scaling via $1/Z_\Phi(p)$ with $\Phi = (h_{\mu\nu}, c_\mu, \bar{c}_\nu)$, which is cancelled by the respective scaling $Z_\Phi^{1/2}$ of each leg in the vertices, for more details see section 5.4. This leads to a canonical momentum running from the propagators in the flow diagrams. What is left is the large momentum scaling of the vertex couplings. For example, the graviton three-point function runs with $p^2 g_3^{1/2}(p) \propto p$ instead the canonical running. This remedies the perturbative divergences and leaves us with a setup with a finite number of divergences and hence counter terms.

# 4 Spectral functional renormalisation group flows

One of our main motivations for using the CS regulator is that Lorentz invariance and the existence of spectral representations are manifest in the flow, see the discussion in section 2.2. We exploit in particular the latter property for defining spectral, Lorentz invariant fRG flows in real time, based on the CS flow (39).

In section 4.1 and section 4.2 we give a brief overview on the spectral representation of correlation functions in quantum field theories, including sum rules for single particle spectral functions and their asymptotic behaviour that are direct consequences of the existence of a spectral representation. In section 4.3 we show how finite flows are computed in practice, allowing for symmetry-preserving functional flows, including gauge-consistent flows. For convenience, in this section we shall mostly use a scalar theory in our discussions.

## 4.1 Spectral representation

The basic ingredient of spectral fRG flows are the spectral representations of the correlation functions, and foremost the Källén-Lehmann (KL) spectral representation of the propagator,

$$G_k(p) = \int_{-\infty}^{\infty} \frac{d\lambda}{2\pi} \frac{\lambda \rho_k(\lambda, \vec{p})}{\lambda^2 + p_0^2}, \tag{42}$$

with $\rho_k(-\lambda, \vec{p}) = -\rho_k(\lambda, \vec{p})$ and

$$\rho_k(\omega, \vec{p}) = 2 \operatorname{Im} G_k(-i\omega_+, \vec{p}), \tag{43}$$

where $\omega_+ = \omega + i0^+$ is the retarded limit. We emphasise that the spectral function is always defined with (43) but the relation (42) does not always hold. As discussed in section 2.1 we have to make sure that the spectral representation of correlation functions at $k = 0$ is maintained also for $k \neq 0$ by choosing an appropriate regulator.

For the two-point function of asymptotic states, the spectral function is positive semidefinite and normalised to unity, if the states are normalised, see also the discussion in section 5. In general this is not the case, since (43) and (42) are mere statements about the causal propagation of the associated operator.

We exemplify these statements within a more detailed discussion of the single scalar field $\phi$ in vacuum. Its two-point function $\Gamma^{(2)}(p^2)$ can be parametrised as

$$\Gamma^{(2)}(p^2) = Z_\phi(p^2)\left(p^2 + m_\phi^2\right), \quad \Gamma^{(2)}\left(-m_\phi^2\right) = 0, \tag{44}$$

with the pole mass $m_\phi$. The respective spectral function $\rho_\phi$ admits the parametrisation

$$\rho_\phi(\lambda, \vec{p}) = \rho^{\text{res}}(\lambda, \vec{p}) + \rho^{\text{cont}}(\lambda, \vec{p}), \tag{45}$$

where $\rho^{\text{res}}$ comprises the resonance contributions to the spectral function,

$$\rho^{\text{res}}(\lambda, \vec{p}) = \frac{\pi}{\lambda} \sum_{i \geq 0, \pm} Z_{\phi,i}\, \delta\left(\lambda \pm \sqrt{\vec{p}^{\,2} + m_{\phi,i}^2}\right), \tag{46}$$

where $i$ labels the stable excitations with masses $m_{\phi,i} > m_{\phi,j}$ for $i > j$ with amplitudes $Z_{\phi,i}$. This includes the ground state, $i = 0$, with the amplitude $Z_{\phi,0}$. The latter is simply the inverse of the on-shell wave function,

$$Z_{\phi,0} = 1/Z_\phi(-m_\phi^2), \tag{47}$$

as can be shown by comparing (44) and (45) on-shell at $p^2 = -m_\phi^2$. The contribution $\rho_{\text{cont}}$ contains the contributions of the scattering continuum,

$$\rho^{\text{cont}}(\lambda, \vec{p}) = \theta\left(\lambda^2 - 4\left(\vec{p}^{\,2} + m_\phi^2\right)\right) f_\phi(\lambda, \vec{p}), \tag{48}$$

with $f_\phi(-\lambda, \vec{p}) = -f_\phi(\lambda, \vec{p})$. It is only non-vanishing beyond the on-shell scattering threshold with the momentum $2m_\phi$, since $m_\phi$ is the mass gap of the theory.

The spectral flows with a spectral CS cutoff are derived as flows in a CS limit (30b) of standard momentum shell cutoff flows as described in section 3.2. Naturally, the persistence of the spectral representation (42) in the presence of the momentum shell regulators facilitates the derivation significantly. Hence, the CS limit may be taken with general regulators whose shape functions (7) are only dependent on spatial momenta squared, $x = \vec{p}^{\,2}/k^2$ or rather $x_\Lambda = \vec{p}^{\,2}/\Lambda^2$, see the two examples (30). It is easy to see that such regulators do not spoil the existence of a spectral representation for positive definite shape functions.

For illustration we again consider the classical regularised propagator (9). Its respective scale dependent spectral function for general regulators $r^\phi(x_\Lambda)$ is given by

$$\rho_k(\lambda, \vec{p}^{\,2}) = \frac{\pi}{\lambda} \sum_{\pm} \delta\left(\lambda \pm \sqrt{\vec{p}^{\,2} + m_\phi^2 + k^2\, r^\phi(x_\Lambda)}\right), \tag{49}$$

which can be shown by inserting (49) in (42),

$$G_{\phi,k}(p_0, \vec{p}) = \frac{1}{p_0^2 + \vec{p}^2 + m_\phi^2 + k^2 r^\phi(x_\Lambda)}, \tag{50}$$

for general shape functions $r^\phi(x_\Lambda)$. Hence, as argued before, we may use spatial momentum regulators (6) with shape functions $r^\phi(x_\Lambda)$ and implement the CS limit (30b) in a spectral way. Note that any class of regulator can be chosen for this limit, we either drop Lorentz invariance or the spectral representation as discussed in section 2.2, see fig. 1. The combination is only obtained in the CS limit, which in its finite form (39) has all three properties.

## 4.2  Sum rules

The KL spectral representation (42) links the infrared asymptote for $\lambda \to 0$ and its ultraviolet asymptote for $\lambda \to \infty$ to the IR and UV behaviour of the Euclidean propagator. This also fixes its normalisation. These properties are discussed and verified in detail in [24, 29, 33, 51]. The UV or IR asymptotic behaviour of the dimensionless Euclidean propagator can be parametrised as

$$\hat{G}_\phi\left(p^2 \to \text{UV/IR}\right) = \frac{Z_\phi}{\hat{p}^2} \frac{\hat{p}^\eta}{(\log \hat{p}^2)^\gamma}, \tag{51}$$

with the dimensionless momentum squared $\hat{p}^2 = p^2/m_{\text{gap}}^2$ and some reference scale $m_{\text{gap}}$. In the UV limit one has the parameters $Z_{\phi,\text{UV}}$, $\eta_{\text{UV}}$, $\gamma_{\text{UV}}$, and in the IR $Z_{\phi,\text{IR}}$, $\eta_{\text{IR}}$, $\gamma_{\text{IR}}$. As discussed in section 4.1, the amplitude $Z_\phi$ is the inverse of the wave function of the two-point function (44).

This general asymptotic form of the propagator includes a power behaviour arising from the anomalous dimension $\eta$ besides the canonical power $-2$, as well as a logarithmic dependence, see e.g. [29, 33, 51] for details. For some non-local theories, the propagator shows an exponential decay behaviour [51], which is not taken into account here. With (51) and the spectral representation (42), the UV asymptote of the spectral function reads

$$\lim_{\hat{\omega} \to \infty} \hat{\rho}(\hat{\omega}) = \frac{Z_{\phi,\text{UV}}}{\hat{\omega}^2} \frac{2\hat{\omega}^{\eta_{\text{UV}}}}{(\log \hat{\omega}^2)^{\gamma_{\text{UV}}}} \left(\sin\left[\frac{\pi}{2}\eta_{\text{UV}}\right] - \cos\left[\frac{\pi}{2}\eta_{\text{UV}}\right] \frac{\pi \gamma_{\text{UV}}}{\log \hat{\omega}^2}\right), \tag{52}$$

and the IR asymptote is given by

$$\lim_{\hat{\omega} \to 0} \hat{\rho}(\hat{\omega}) = \frac{Z_{\phi,\text{IR}}}{\hat{\omega}^2} \frac{2\hat{\omega}^{\eta_{\text{IR}}}}{(\log \hat{\omega}^2)^{\gamma_{\text{IR}}}} \left((2 - \eta_{\text{IR}}) + \frac{2\gamma_{\text{IR}}}{\log \hat{\omega}^2}\right). \tag{53}$$

The UV limit already entails that only for $\eta_{\text{UV}} = 0$, $\gamma_{\text{UV}} = 0$ we have a normalisable spectral function with

$$\int_0^\infty d\lambda\, \lambda\, \rho_\phi(\lambda) = Z_{\phi,\text{UV}}, \tag{54a}$$

which is in one-to-one correspondence with the commutation relations $[\phi(t, \vec{x}), \partial_t \phi(t, \vec{y})] = Z_{\phi,\text{UV}} \delta(\vec{x} - \vec{y})$. The standard normalisation is obtained for $Z_{\phi,\text{UV}} = 1$, which entails canonical commutation relations.

In turn, for $\eta_{\mathrm{UV}} < 0$ or $\gamma_{\mathrm{UV}} > 0$ the UV-tail of the spectral function is negative, and the respective field does not describe an asymptotic state. Moreover, the spectral function is normalised to zero,

$$\int_0^\infty d\lambda\, \lambda\, \rho_\phi(\lambda) = 0\,. \tag{54b}$$

In QCD this is the well-known Oehme-Zimmermann super convergence property [52, 53] for the gluon in covariant gauges, for an evaluation in the Landau gauge see [33]. In asymptotically safe gravity it holds true for the background graviton, for a reconstruction see [54].

For $\eta_{\mathrm{UV}} > 0$ or $\gamma_{\mathrm{UV}} < 0$ the UV tail of the spectral function is positive, but the spectral function is not normalisable,

$$\lim_{\Lambda \to \infty} \int_0^\Lambda d\lambda\, \lambda\, \rho_\phi(\lambda) \to \infty\,, \tag{54c}$$

in the absence of IR singularities. Equation (54c) holds true for the spectral function of the fluctuation graviton in covariant gauges, see [54] for a reconstruction, and [24] for a direct computation with the spectral fRG. Note that also in this case the field does not generate an asymptotic state by applying it to the vacuum, $\phi|0\rangle$. However, this is not to be expected in a non-Abelian gauge theory or quantum gravity.

### 4.3 Spectral renormalisation, symmetries

It has been discussed in [28], how the momentum integrals of fully non-perturbative loop integrals can be computed within dimensional regularisation. It has also been shown, how a fully gauge-consistent functional renormalisation scheme can be set up by also applying *spectral dimensional regularisation*. One also can use a Bogoliubov-Parasiuk-Hepp-Zimmermann–type (BPHZ) subtraction scheme, *spectral* BPHZ-*regularisation*. For details we refer the reader to this work, here we only briefly recapitulate the important properties of spectral renormalisation.

The spectral renormalisation scheme in [28] has been set up for general functional approaches, and has been exemplified within the Dyson-Schwinger equation (DSE) for the scalar theory. The respective loop equations contain up to two-loop diagrams with non-perturbative propagators and vertex functions. In the present case of the spectral fRG we only have to consider the renormalisation of one-loop diagrams which facilitates the task. One of the lines carries the cutoff insertion, and the momentum routing is typically chosen such that it only depends on the loop momentum $q$. In terms of the frequency dependence, the line with the cutoff insertion simply leads to two classical propagators with the spectral masses $\lambda_1^2$ and $\lambda_2^2$, both carrying the loop frequency $q_0$. The CS or spatial regulator does not depend on the loop frequency, but only on $x = \vec{q}^{\,2}/k^2$. To facilitate numerical computations in $d > 1$, it is advantageous to use a spectral representation of the full regulator line or more precisely the propagator squared,

$$G_\phi(q)\partial_t R^\phi(x) G_\phi(q) = \frac{\partial_t R_\phi(x)}{q^2} \int_{-\infty}^\infty \frac{\mathrm{d}\lambda}{2\pi} \frac{\lambda\, \rho_{G^2}(\lambda)}{\lambda^2 + q^2}\,. \tag{55}$$

In (55) we have used, that Lorentz invariance allows us to reduce $\rho_i(\lambda, \vec{q})$ to $\rho_i(\lambda) = \rho_i(\lambda, 0)$ within spectral representations such as (42) and (55),

$$\int_{-\infty}^\infty \frac{d\lambda}{2\pi} \frac{\lambda \rho_i(\lambda, \vec{q})}{\lambda^2 + q_0^2} = \int_{-\infty}^\infty \frac{d\lambda}{2\pi} \frac{\lambda \rho(\lambda, 0)}{\lambda^2 + q^2}\,. \tag{56}$$

Note that the regulator derivative in (55) is simply multiplying the spectral representation of $G^2$. This is a consequence of the regulator not carrying the loop frequency. The spectral density $\rho_{G^2}$ in (55) is defined as

$$\rho_{G^2}(\omega) = 2\operatorname{Im}\left[\omega_+^2 G(\omega_+)^2\right]. \tag{57}$$

We may either use (55) or the product of the two spectral functions for the propagators on the right hand side of (55). In both cases, general flow digrams Diag(**p**) of the flow of vertex functions and inverse propagators with the external momenta $\mathbf{p} = (p_1, ..., p_n)$ have the representation

$$\text{Diag}(\mathbf{p}) = \int \frac{d^d q}{(2\pi)^d} \text{Vert}(\mathbf{l}, \mathbf{p}) \prod_{i=1}^{N_{\max}} \int_{-\infty}^{\infty} \frac{d\lambda_i}{2\pi} \frac{\lambda_i \rho_i(\lambda_i, \vec{l}_i)}{\lambda_i^2 + (l_i)_0^2}, \tag{58}$$

where $\mathbf{l} = (q, q + p_1, ...., )$ is the vector of all momenta entering the propagators and vertices of the loop diagram at hand, and $N_{\max}$ is the number of spectral functions. The factor $\text{Vert}(\mathbf{l}, \mathbf{p})$ stands for the momentum dependences of vertex and regulator factors and possible projections and is a rational function in the momenta $\mathbf{l}$ and $\mathbf{p}$.

For example, for constant vertex functions and using (55), $\text{Vert}(\mathbf{l}, \mathbf{p}) \propto \frac{1}{q_0^2}$, and $N_{\text{Max}}$ is simply the number of internal lines including the regulator line. Then, the $\rho_i$ are the spectral functions of the fields $\phi_i$ propagating in the respective line and $\rho_1 = \rho_{G^2}$. In turn, if only using the spectral representation of the propagators, the vertex factor $\text{Vert}(\mathbf{l}, \mathbf{p})$ has no frequency and momentum dependence, but $N_{\text{Max}} \to N_{\text{Max}} + 1$: it is the number of internal lines and the regulator line counts twice.

With (56), the momentum integral in (58) has the standard form of a one loop perturbative integral, and can be computed with dimensional regularisation with $d \to d - 2\epsilon$ and $\epsilon \to 0$. We are led to

$$\text{Diag}(\mathbf{p}) = \prod_{i=1}^{N_{\max}} \int_{-\infty}^{\infty} \frac{d\lambda_i}{2\pi} \lambda_i \rho_i(\lambda_i, 0) F_{\text{diag}}(\boldsymbol{\lambda}, \mathbf{p}; \epsilon), \tag{59}$$

with

$$F_{\text{diag}}(\boldsymbol{\lambda}, \mathbf{p}; \epsilon) = \int \frac{d^d q}{(2\pi)^d} \text{Vert}(\mathbf{l}, \mathbf{p}) \prod_{i=1}^{N_{\max}} \frac{1}{\lambda_i^2 + (l_i)^2}. \tag{60}$$

Equation (59) has the same form as the general spectral integrals considered in [28] and hence is treated the same way. Here we briefly recapitulate the main results obtained there and refer the reader to [28] for more details.

To begin with, for power-counting divergent perturbative momentum integrals, $F_{\text{diag}}$ contains $1/\epsilon$-terms in even dimensions $d = 2n$ with $n \in \mathbb{N}$. It is tempting to apply the minimal subtraction idea of only subtracting these divergent pieces. This would amount to simply dropping the $1/\epsilon$-terms in $F_{\text{diag}}$. However, as thoroughly discussed in [28], the remaining spectral integrations have the same ultraviolet degree of divergence and may not be finite. Note that these divergences are sub-divergences and are absent at one loop perturbation theory where the spectral functions are Dirac $\delta$-functions. This leaves us with two choices:

(i) *Spectral dimensional renormalisation:* if we want to maintain all symmetry-features of dimensional regularisation, we also have to perform the UV part of the spectral integrations analytically. This can be done using splits

$$\rho(\lambda, \vec{q}) = \rho_{\text{IR}}(\lambda, \vec{q}) + \rho_{\text{UV,an}}(\lambda, \vec{q}), \tag{61}$$

where the 'IR' part decays sufficiently fast for large spectral values, and $\rho_{\mathrm{UV,an}}$ carries the UV-tail of the spectral function and its form is chosen such that it facilitates the analytic computation of the UV-part of the spectral integrations. Finally, we are left with $1/\epsilon$ terms from both the momentum and spectral integrals, which can be subtracted by an appropriate choice of $\partial_t S_{\mathrm{cl}}$ in (39).

(ii) *Spectral* BPHZ-*renormalisation:* We implement the RG-conditions at an RG-scale $\mu$ in terms of subtractions at the level of the integrand in (60). This amounts to subtracting a Taylor expansion in $\mathbf{p}$ of $F_{\mathrm{diag}}$. For the sake of simplicity we restrict ourselves to a case with one external momentum and a quadratic divergence, e.g. the flow of the two-point function $\Gamma^{(2)}(p)$ in a scalar theory in $d = 4$ dimensions. Then, $\mathbf{p} = p$ and the BPHZ-subtraction reads schematically,

$$\mathrm{Diag}_{\mathrm{ren}}(\mathbf{p}) = \prod_{i=1}^{N_{\mathrm{max}}} \int_{-\infty}^{\infty} \frac{d\lambda_i}{2\pi} \lambda_i \rho_i(\lambda_i, 0) \Bigg[ F_{\mathrm{diag}}(\boldsymbol{\lambda}, p; \epsilon)$$
$$- F_{\mathrm{diag}}(\boldsymbol{\lambda}, \mu; \epsilon) - (p^2 - \mu^2) \left. \frac{\partial F_{\mathrm{diag}}(\boldsymbol{\lambda}, p; \epsilon)}{\partial p^2} \right|_{p^2 = \mu^2} \Bigg]. \tag{62}$$

In (62) we can take the limit $\epsilon \to 0$ before performing the spectral integrations which are manifestly finite. The showcase (62) straightforwardly extends to the flow of general correlation functions with the standard BPHZ-procedure. Evidently, the subtraction terms constitute a specific choice of $\partial_t S_{\mathrm{cl}}$ in (39).

This closes our brief recapitulation of the conceptual results in [28], and the discussion of their application to the spectral CS-flows: The spectral dimensional or BPHZ-renormalisation is implemented by a respective choice of the flow of the counter term action $\partial_t S_{\mathrm{cl}}$ in (39). This leads us to manifestly finite spectral flows within a systematic flowing renormalisation scheme.

Evidently, the spectral BPHZ-renormalisation is technically less challenging, and is the renormalisation method of choice in most cases. However, we emphasise that the $\epsilon \to 0$-limit and the integration do not commute, and hence the spectral BPHZ-renormalisation and the spectral dimensional renormalisation may not agree in terms of symmetries. This may be specifically important for gauge theories. Either way this allows us to define *finite* spectral flows.

# 5 Spectral renormalisation at work

In this section we discuss the choice and implementation of renormalisation group conditions and their flows, adapted to the theory class or physics situation at hand. We start with the simple example of a $\phi^4$ theory in $d \leq 4$ dimensions in section 5.1. This and the following example of a Yukawa model in section 5.2 allow us to discuss the generic setting of flowing spectral RG-conditions. Asymptotically free non-abelian gauge theories are discussed in section 5.3, specifically concentrating on infrared and ultraviolet asymptotes of the spectral function. Finally we discuss asymptotically safe quantum gravity in section 5.4, again concentrating on the ultraviolet and infrared asymptotes, as well as the consequences of a non-trivial ultraviolet momentum scaling for renormalisation.

## 5.1 Spectral renormalisation in scalar theories

Here, we discuss spectral renormalisation in the $\phi^4$-theory in $d = 3, 4$ dimensions. The classical action is defined in (3), and we recall it here for the sake of convenience,

$$S_\phi[\phi] = \int_x \left[ \frac{1}{2} \phi \left( -\partial^2 + m_\phi^2 \right) \phi + \frac{\lambda_\phi}{4!} \phi^4 \right]. \tag{63}$$

A typical spectral function can be parametrised as

$$\rho(\lambda, \vec{p}) = Z_\phi \, \delta\left(\lambda^2 - \vec{p}^{\,2} - m_\phi^2\right) + \tilde{\rho}(\lambda, \vec{p}). \tag{64}$$

Employing the split of (45), the delta pole term in (64) at the one particle pole mass $m_\phi^2$ is identified with the resonance contribution, while $\tilde{\rho}$ corresponds to the continuum part. In the case of multi-particle bound states, the resonance contribution in (45) will contain further delta pole contributions in addition to the one-particle pole in (64). Such bound states are for example observed in $d = 3$ dimensions close to the phase transition from the ordered into the disordered phase [55–57]. For the sake of simplicity, we restrict ourselves to the case of a single particle pole here.

The spectral function is antisymmetric, i.e. $\rho(-\lambda, \vec{p}) = -\rho(\lambda, \vec{p})$, which also holds for both terms separately in (64). The part $\tilde{\rho}$ in (64) encodes the scattering spectrum of the theory. In the presence of a non-zero background it has support for $\lambda^2 > 4(\vec{p}^{\,2} + m_\phi^2)$. $\tilde{\rho}$ also carries higher scattering thresholds. The general sum rule (54a) imposes the constraint

$$Z_\phi + \int_{-\infty}^{\infty} \frac{d\lambda}{\pi} \tilde{\rho}(\lambda, \vec{p}) = Z_{\text{UV}}, \tag{65}$$

where $Z_{\text{UV}} = \lim_{p\to\infty} 1/Z(p)$. Generally, the propagator can be parametrised as

$$G_\phi(p) = \frac{1}{Z(p)} \frac{1}{p^2 + m_\phi^2}, \tag{66}$$

that is the inverse of (44). As also discussed there, the pole amplitude or residue $Z_\phi$ in (64) is related to the wave function $Z(p)$ in (66) with $Z_\phi = 1/Z(p^2 = -m_\phi^2)$. Equation (64) elucidates the relation of the parameters in the spectral function to the fundamental parameters of the theory. In general dimensions $d \leq 4$ both $Z_\phi$ (in $d = 4$) and $m_\phi$ (in $d \geq 2$) are subject to renormalisation. The resonances $m_i$ with their respective residues $Z_i$ as well as the scattering continuum $\tilde{\rho}$ carry the dynamics, and are comprised in the momentum dependence of $Z(p^2)$.

We start our analysis of the spectral renormalisation with the $\phi^4$ theory in $d = 3$, where the theory is super-renormalisable and the only UV divergence is a logarithmic one in the mass. Consequently, the Callan-Symanzik flow of the theory is manifestly finite already without the counter terms.

Turning towards the flow of the inverse propagator, we discuss some of its aspects at the example of the tadpole diagram. Evaluated at vanishing field expectation $\phi = 0$, the tadpole is the only diagram present, hence the full flow reads

$$\partial_t \Gamma_k^{(2)}(p) = k^2 \int \frac{d^3q}{(2\pi)^3} \Gamma_k^{(4)}(p, -p, q, -q) G_\phi(q)^2. \tag{67}$$

By means of Jordan's lemma, we can choose to integrate over the Euclidean or Minkowski domain. The external momentum is in general complex, $p^2 \in \mathbb{C}$. We use $p^2 \geq 0$ for the Euclidean (or spacelike) and $p^2 \leq 0$ for the Minkowski (timelike) branch.

Assuming that the propagator decays with $1/q^2$ and the full four-point function stays finite for large momenta, the loop momentum integral on the right hand side of (67) converges.

Trivially, former assumptions hold true for the classical propagator and four-point function. Therefore $\partial_t \Gamma^{(2)}(p)$ stays finite for $p^2 \to \infty$, and hence the flow sustains the assumed ultraviolet behaviour of the propagator. Finiteness for $p^2, q^2 \to \infty$ can also be shown for the four-point function. Hence the argument holds true for the full theory, and no regularisation is required in the scalar $\phi^4$-theory in $d = 3$.

In three dimensions the counter term flow $\partial_t S_{\text{ct}}$ is not required for having a finite flow, it is only needed for satisfying specific RG conditions. Without the counter term flow, the RG conditions change with the flow-induced changes of $Z_\phi$ and $m_\phi$. This is the common approach in functional flows where the RG conditions are implicitly specified by the choice of the effective action at the initial cutoff scale $k_{\text{init}}$, and freely evolve with the flow.

Let us now focus on the flow of the pole mass $m_\phi$. With (36) and $\mathcal{D}_k \propto \Lambda(k)$ the corresponding flow of the counter term action reads

$$\partial_t S_{\text{ct}} = \frac{1}{2} \int_x \Delta \dot{m}_k^2 \phi^2 \,. \tag{68}$$

Specifying the RG conditions for all $k$ and hence their flows, fixes $\Delta \dot{m}_k^2$ in (68). Here, we discuss this at the example of the on-shell RG-conditions in (40) and their flows (41). The mass renormalisation condition entails that at each flow step, the pole mass is given by

$$m_\phi^2 = k^2 \,, \tag{69}$$

and its flow is given by

$$\dot{m}_\phi = -2k^2 \,. \tag{70}$$

Including the counter term action (68), the full flow of the two point function schematically reads

$$\partial_t \Gamma_k^{(2)} = \text{Flow}^{(2)} - \Delta \dot{m}_k^2 \,, \tag{71}$$

where $\text{Flow}^{(2)}$ represents the contributions from the diagrams. We emphasise again that the flow in (71) is manifestly finite. In $d = 3$ this trivially holds true since $\text{Flow}^{(2)}$ is finite by itself. In $d = 4$ this is not the case. There, $\Delta \dot{m}_k^2$ acts as a genuine counter term cancelling the logarithmic UV divergence of the mass flow.

Equation (70) fixes the mass counter term flow,

$$\Delta \dot{m}_k^2 = -2k^2 - \text{Flow}^{(2)}\big|_{p^2 = -k^2} \,. \tag{72}$$

Evidently, the flow of the RG-condition does not invoke any fine-tuning problem, since (72) can be evaluated at each flow step. The initial condition for (71) is simply given by

$$\lim_{k \to \infty} \Gamma_k^{(2)} = p^2 \,. \tag{73}$$

This implies $\lim_{k \to \infty} Z_\phi \to 1$, as the scattering continuum for the scalar theory with an infinitely heavy scalar vanishes.

In the present case, $Z_\phi$ does not carry the renormalisation of the wave function, but rather the normalisation of the spectral function. In $d = 3$ dimensions the full propagator has the UV limit

$$G_\phi(p^2) \overset{p^2 \to \infty}{\longrightarrow} \frac{1}{p^2} \,, \tag{74}$$

and hence the spectral function satisfies (54a) with $Z_{\mathrm{UV}} = 1$. This implies that $Z_\phi \neq 1$, as both conditions together would entail, that no scattering continuum is present. This already suggests that the second RG condition in (40) should not be implemented for physical fields.

In $d = 4$ dimensions the $\phi^4$-theory requires the renormalisation of $Z_\phi, m_\phi, \lambda_\phi$. Still, the CS flow only shows a (logarithmic) divergence for the tadpole term, and the flows for $Z_\phi$ and $\lambda_\phi$ are manifestly finite. The flow of the renormalisation of the mass is still given by (72), which holds true in general dimensions.

We may or may not accompany this condition with the RG-conditions for $Z_\phi$ and $\lambda_\phi$ in (40). This is optional, as the CS flow is already finite after using (72). If we do not enforce the RG-conditions, they will dynamically change during the flow. As already discussed above, this is the standard procedure in momentum cutoff flows, where the renormalisation conditions at $k = 0$ are only implicitly encoded in the initial conditions at $k \to \infty$.

## 5.2 Spectral renormalisation in Yukawa theories

Yukawa theories describe many exciting physics phenomena, most notably fermionic systems with bosonic bound states or resonances ranging from non-relativistic ultracold systems with fermionic atoms and molecules/pairs, over superconducting systems with Cooper pairs to relativistic quark–meson-diquark systems in QCD. Moreover, the Higgs sector constitutes a pivotal part of the Standard Model of Particle Physics.

In the present conceptual work, we consider a simple relativistic theory with one fermion flavour $\psi$ and a scalar boson $\phi$ in $d = 4$ dimensions with the classical action $S_{\mathrm{yuk}}[\phi]$,

$$S_{\mathrm{yuk}}[\Phi] = S_\phi[\phi] + \int_x \left\{ \bar\psi \left( \gamma_\mu \partial_\mu + m_\psi \right) \psi + h_\phi \, \bar\psi \, \phi \, \psi \right\}, \tag{75}$$

and the superfield

$$\Phi = (\psi, \bar\psi, \phi). \tag{76}$$

$h_\phi$ is the Yukawa coupling and $S_\phi[\phi]$ is the same as in the previous example, Equation (63). Its CS flow is also discussed in LPA in appendix A. We concentrate on structural aspects of mixed boson-fermion flows, and the present results readily extend to general systems of this type, naturally including purely fermionic ones.

Before we discuss general aspects of CS flows of the Yukawa theory, we would like to comment on the role of a finite chemical potential $\mu_\psi$ for the fermions,

$$\int_x \bar\psi \, \gamma_\mu \partial_\mu \psi \to \int_x \bar\psi \left[ \gamma_0 (\partial_0 + \mu_\psi) + \gamma_i \partial_i \right] \psi. \tag{77}$$

In momentum space the chemical potential simply shifts the fermionic frequency $p_0$ by $-i\mu_\psi$. This shift entails that correlation functions in the above theory only depend on $\tilde p = (\tilde p_0, \vec p)$ with $\tilde p_0 = p_0 - i\mu_\psi$ for $\mu_\psi < m_\psi$. For a proof in the present fRG setup, see [50, 58] and, for further discussions, we refer to Ref. [59]. This property is the Silver-Blaze property [60], originally derived as the $\mu_\psi$-independence of observables below the mass threshold. In any case, if not stated otherwise, we shall focus on the limit $\mu_\psi \to 0$ in our discussion from here on.

In $d \leq 4$ dimensions, the Yukawa theory with the classical action (75) is renormalisable. In four dimensions it has the relevant parameters $m_\psi, m_\phi, h_\phi, \lambda_\phi$ and the wave function renormalisations $Z_\phi, Z_\psi$ of the fields. In $d = 3$ dimensions, only the masses $m_\phi, m_\psi$ and the wave function renormalisation $Z_\psi$ of the fermion require renormalisation, while the scalar mass is the only renormalised parameter left in $d = 2$ dimensions.

We also remark that in the pure Yukawa theory, that is in the absence of any scalar self-interaction term in $S_\phi$, we can eliminate the fermionic mass at the expense of a linear (source) term in the scalar field which couples only to its constant mode $\int_x \phi$. This is typically done in QCD, where the scalar (mesonic) fields are dynamical low energy degrees of freedom in the first place, see, e.g., [4,61]. Applying the shift

$$\phi \rightarrow \phi - \frac{m_\psi}{h_\phi}, \tag{78a}$$

to the classical action (75), we arrive at

$$\frac{1}{2}\int_x \phi\left(-\partial^2 + m_\phi^2\right)\phi + \int_x \bar{\psi}\left(m_\psi + h_\phi\,\phi\right)\psi \rightarrow \frac{1}{2}\int_x \phi\left(-\partial^2 + m_\phi^2\right)\phi + h_\phi\int_x \bar{\psi}\,\phi\,\psi - c_\phi\int_x \phi\,, \tag{78b}$$

with the constant source

$$c_\phi = \frac{m_\phi^2\, m_\psi}{h_\phi}\,. \tag{78c}$$

Note that we have dropped field-independent terms in (78b). After this transformation, we are left with an action without fermionic mass parameter, which is also not regenerated by quantum fluctuations. The fermionic mass parameter is inessential as defined in [46,47]. The explicit chiral symmetry breaking due to the fermionic mass term in (75) can be implemented as a linear "tilt" of the effective action which can be added after fluctuations have been integrated out. Note that this "tilt" $c_\phi$ does not require renormalisation [62]. In practice, we can therefore set this parameter to zero and consider only a theory which is invariant under chiral symmetry transformations $\psi \rightarrow \exp(i\gamma_5)\psi$ and $\bar{\psi} \rightarrow \bar{\psi}\exp(i\gamma_5)$ (and correspondingly for the scalar field). We emphasise that the number of relevant parameters has not been changed by this transformation, as $m_\psi \sim c_\phi$.

### 5.2.1 Spectral fermionic regulators

The spectral fRG is based on the spectral representation of the propagators, which is presented in (64) for the scalar field, and the spectral CS cutoff is implemented with (69). It respects all symmetries of the scalar theory and naturally we would like to augment the scalar spectral regulator with a spectral regularisation of the fermion that preserves all symmetries of the theory. This includes internal symmetries such as the chiral symmetry as present in (75) for $m_\psi = 0$. A class of regulators which respects chiral symmetry is given by

$$R_d^\psi = Z_\psi\, i\epsilon\, \bar{R}_d^\psi\,, \quad \text{with} \quad \bar{R}_d^\psi = k\, r_d^\psi(x)\,, \tag{79}$$

with $\epsilon$ being

$$\epsilon(p) = \frac{\slashed{p}}{p}\,, \qquad \text{or} \qquad \epsilon(\vec{p}) = \frac{\vec{\gamma}\cdot\vec{p}}{\sqrt{\vec{p}^2}}\,. \tag{80}$$

In either case, we have $\epsilon^2 = \mathbb{1}$. While $\bar{\psi}\epsilon(p)\bar{R}_d^\psi\psi$ is a Lorentz-invariant object, it may not admit a spectral representation. In turn, the operator $\bar{\psi}\epsilon(\vec{p})\bar{R}_d^\psi\psi$ is not invariant under Lorentz transformations, but is more likely to sustain the spectral representation. In (79), we have separated the Dirac tensor structure and the RG running $Z_\psi i\epsilon$ of the regulator from the (scalar) regularisation part $\bar{R}_d^\psi$.

A class of fully Lorentz-invariant regulators that does not violate the spectral representation, but violates chiral symmetry, is given by regulators with the scalar tensor structure,

$$R_s^\psi = Z_\psi \bar{R}_s^\psi, \quad \text{with} \quad \bar{R}_s^\psi = k\, r_s^\psi(x), \tag{81}$$

with the same factorisation of (scalar) tensor structure and RG running from the regularisation part $\bar{R}_s^\psi$.

In the following we consider a combination of the Lorentz-invariant sum of regulators $R^\psi = R_d^\psi + R_s^\psi$ with $\epsilon(p)$. The general case with spatial regulators with $\epsilon = \epsilon(\vec{p})$ and/or shape functions $r(\vec{p}^2/k^2)$ require different dressings for tensor structure parallel and transverse to the $p_0$ direction, similar to thermal or density splits. This case is deferred to appendix B.

In the Lorentz-invariant case the full fermionic two-point function is parametrised as

$$\Gamma_{\psi\bar\psi}^{(2)} = Z_\psi(p)\Big[\mathrm{i}\slashed{p} + M_\psi(p)\Big], \tag{82}$$

with the wave function $Z_\psi(p)$ and the mass function $M_\psi(p)$. In contradistinction to the scalar theory discussed before the fermionic theory allows for a unique projection onto the mass function with $M_\psi(p) = 1/Z_\psi(p)\mathrm{tr}_d\, \Gamma_{\psi\bar\psi}^{(2)}$, where $\mathrm{tr}_d$ is the Dirac trace. Then, for asymptotic states the pole condition is given by

$$M_\psi^2\left(p^2 = -m_{\psi,\mathrm{pole}}^2\right) = m_{\psi,\mathrm{pole}}^2. \tag{83}$$

The chiral properties of these different regulator classes are best discussed with the full fermionic propagator $G_\psi$ as given by (2) and (82), related to a sum of chiral and scalar regulators. This leads

$$G_\psi(p) = -\mathrm{i}\slashed{p}\, G_\psi^{(d)}(p) + M_\psi\, G_\psi^{(s)}(p), \tag{84}$$

with the convenient normalisation $M_\psi = M_\psi(0)$ and the regularised Dirac (d) and scalar (s) parts of the propagator. The Dirac dressing is given by

$$G_\psi^{(d)}(p) = \left[1 + A_\psi(p)\frac{1}{p}\bar{R}_d^\psi(p)\right]G_\psi^{(u)}, \tag{85}$$

where the $1/p$ part in the numerator stems from the chiral infrared regularisation and

$$A_\psi(p) = \frac{Z_\psi}{Z_\psi(p)}. \tag{86}$$

The scalar dressing reads

$$G_\psi^{(s)}(p) = \frac{1}{M_\psi}\left[M_\psi(p) + A_\psi(p)\bar{R}_s(p)\right]G_\psi^{(u)}. \tag{87}$$

Both are proportional to the universal part

$$G_\psi^{(u)}(p) = \frac{1}{Z_\psi(p)}\frac{1}{\left(p + A_\psi \bar{R}_d^\psi\right)^2 + \left(M_\psi(p) + A_\psi \bar{R}_s^\psi\right)^2}, \tag{88}$$

with all the properties of a scalar propagator. We emphasise that $G_\psi^{(u)}(p)$ admits a spectral representation with suitably chosen regulators.

### 5.2.2 Chiral CS regulators

We proceed with a detailed discussion of the possible choices, starting with the chiral CS cutoff. In terms of the shape functions this is given by

$$\left(r_d^{\psi}(x), r_s^{\psi}(x)\right) = (1, 0). \tag{89}$$

This leads us to (84) with

$$G_{\psi}^{(d)}(p) = \left[1 + A_{\psi}(p)\frac{k}{p}\right]G_{\psi}^{(u)},$$

$$G_{\psi}^{(s)}(p) = M_{\psi}(p)G_{\psi}^{(u)}, \tag{90}$$

with the universal part (88) being reduced to

$$G_{\psi}^{(u)}(p) = \frac{1}{Z_{\psi}(p)}\frac{1}{\left(p + A_{\psi}(p)k\right)^2 + M_{\psi}(p)^2}. \tag{91}$$

Let us now investigate the complex structure of the universal part of the classical propagator in the presence of the chiral CS cutoff. In the classical limit, (91) is simply given by

$$G_{\psi,\mathrm{cl}}^{(u)}(p) = \frac{1}{(p + k)^2 + M_{\mathrm{cl}}^2}, \tag{92}$$

where $p = \sqrt{p^2} = \sqrt{p_0^2 + \vec{p}^{\,2}}$ and $M_{\mathrm{cl}}$ is the (classical) fermionic mass. The pole positions are easily extracted and read as follows for $\vec{p} = 0$:

$$p_0 = -k \pm \mathrm{i}\, M_{\mathrm{cl}}. \tag{93}$$

Thus, chiral CS regulators lead to complex conjugate poles that invalidate the spectral representation. As shown in [63], such poles are likely to propagate through the systems of coupled functional equations leading to the lack of a spectral representation for all fields as well as triggering further cuts and poles. We also remark that the occurrence of $\sqrt{p^2}$ leads to further cuts in the presence of a chemical potential, where we have $\sqrt{p^2} \to \sqrt{\tilde{p}^2}$. The latter expression leads to a cut in the complex plane starting at $\tilde{p} = 0$ which invalidates the Silver-Blaze property.

This leads us to the important conclusion that neither the chiral CS cutoff nor momentum-dependent chiral regulators are well-suited for spectral considerations and non-vanishing chemical potential, precisely due to their chiral nature.

### 5.2.3 CS regulator

The other natural choice is the standard CS cutoff,

$$(r_d^{\psi}(x), r_s^{\psi}(x)) = (0, 1). \tag{94}$$

This leads us to (84) with

$$G_{\psi}^{(d)}(p) = G_{\psi}^{(u)},$$

$$G_{\psi}^{(s)}(p) = \frac{M_{\psi}(p)}{M_{\psi}}\left[1 + \frac{A_{\psi}(p)}{M_{\psi}(p)}k\right]G_{\psi}^{(u)}, \tag{95}$$

with the universal part (88) being reduced to

$$G_{\psi}^{(u)}(p) = \frac{1}{Z_{\psi}(p)}\frac{1}{p^2 + (M_{\psi}(p) + A_{\psi}(p)k)^2}. \tag{96}$$

The standard CS regulator simply shifts the fermionic mass function $M_\psi(p)$ by $A_\psi(p)k$. As in the case of the chiral CS regulator, we analyse the complex structure at the example of the classical propagator. Then, the universal part (96) reduces to

$$G_\psi^{(u)}(p) = \frac{1}{p^2 + (M_{\text{cl}} + k)^2}\,, \tag{97}$$

with the pole positions

$$p_0 = \pm i(M_{\text{cl}} + k)\,, \tag{98}$$

at Minkowski frequencies. As in the scalar case, the regulator simply shifts the pole position by $k$ and the flow is one of a theory of massive fermions. The chiral limit is approached for $k \to 0$ in a controlled way.

For the standard CS cutoff defined in (94), the spectral representation of the full propagator is given by

$$G_\psi(p) = -i\slashed{p} \int_{-\infty}^\infty \frac{d\lambda}{2\pi} \frac{\lambda \rho_{\psi,k}^{(d)}(\lambda, \vec{p}^2)}{\lambda^2 + p_0^2} + M_\psi \int_{-\infty}^\infty \frac{d\lambda}{2\pi} \frac{\lambda \rho_{\psi,k}^{(s)}(\lambda, \vec{p}^2)}{\lambda^2 + p_0^2}\,. \tag{99}$$

Here, $\rho_\psi^{(d/s)}$ are the spectral functions associated with the scalar parts of $G_\psi^{(d/s)}$. We shall parametrise the spectral functions as follows:

$$\rho^{(d/s)}(\lambda, \vec{p}) = Z_\psi \, \delta(\lambda^2 - \vec{p}^2 - m_{\psi,\text{pole}}^2) + \tilde{\rho}^{(d/s)}(\lambda, \vec{p})\,, \tag{100}$$

with $\tilde{\rho}^{(d/s)}(-\lambda, \vec{p}) = -\tilde{\rho}^{(d/s)}(\lambda, \vec{p})$. The pole mass $m_{\psi,\text{pole}}$ increases with $k$ which leads to the regularisation of the theory.

Similarly to the cutoff propagator of the scalar theory, there is a spectral representation for that of the fermionic cutoff propagator with the standard CS cutoff. Before we present this spectral representation, however, we note that the fermionic cutoff propagator reads

$$G_\psi(q)\partial_t R_s^\psi(q)G_\psi(q) = G_\psi(q)Z_\psi \, k\left[(1 - 2\gamma_\psi)\, r_s^\psi(x) + \partial_t r_s^\psi(x)\right]G_\psi(q)\,. \tag{101}$$

This reduces to

$$\left[G_\psi \partial_t R_s^\psi G_\psi\right](q) = G_\psi(q)\left[Z_\psi k(1 - 2\gamma_\psi)\right]G_\psi(q)\,, \tag{102}$$

for the standard CS cutoff with $r_s^\psi(x) = 1$. As already indicated above, the cutoff propagator has a spectral representation:

$$\left[G_\psi(q)\partial_t R_s^\psi(q)G_\psi\right](q) = -i\slashed{q} \int_{-\infty}^\infty \frac{d\lambda}{2\pi} \frac{\lambda \rho_{\psi,\text{reg}}^{(d)}(\lambda, \vec{q})}{q_0^2(\lambda^2 + q_0^2)} + M_q \int_{-\infty}^\infty \frac{d\lambda}{2\pi} \frac{\lambda \rho_{\psi,\text{reg}}^{(s)}(\lambda, \vec{q})}{q_0^2(\lambda^2 + q_0^2)}\,. \tag{103}$$

With the spectral representations for the fermionic propagators and the fermionic cutoff propagator, the flow diagrams for mixed scalar-fermionic theories have the momentum and spectral structure displayed in (58) with its renormalised form (62).

### 5.2.4 Shift symmetry for the scalar field

Mixed fermion-boson theories such as the standard Yukawa theory with the action (75) allow for some remarkable and powerful reparametrisations. The common one is given by (78), which removes the explicit fermion mass term from the theory in favour of a linear term in

the $\phi$ field. However, this structure is more generic. Here we elucidate this property with the scalar Yukawa theory. The vector Yukawa theory is discussed in section 5.2.5.

In (78) the shift of the scalar field was used to remove the fermionic mass term at the expense of an explicit linear breaking term in the scalar field. Evidently, this shift can also be used to remove the standard fermionic CS regulator which is nothing but a mass term. Moreover, we can formulate this shift in momentum space with

$$\phi(p) \to \phi(p) - \frac{1}{h_\phi}\big[m_\psi + h_\phi R_s^\psi(p)\big]. \tag{104}$$

After the shift with (104) only the coefficient $c_\phi$ of the linear scalar term depends on $R_\psi$ with the general form

$$c_\phi(p) = \frac{m_\phi^2}{h_\phi}\big[m_\psi + h_\phi R_s^\psi(p)\big]. \tag{105}$$

Equation (105) entails, that a massive Yukawa theory in the presence of a scalar momentum regulator for the fermions is identical to a chiral Yukawa theory without a cutoff for the fermions, but a momentum-dependent linear breaking term. Importantly, the flow equation does not include a fermion loop as we have the identity

$$\frac{\delta}{\delta R_s^\psi}\Big[\Gamma_k[\Phi] + \int c_\phi \phi\Big] = 0. \tag{106}$$

Put differently, the Yukawa theory includes all fermion dispersions in the scalar effective potential, or rather a given momentum dependent scalar field background. Via a Legendre transform this entails a space-time dependent source coupled to the scalar field.

Interestingly, we may still proceed with the fermionic regulator despite the identity (106): the EoM for the scalar field in the shifted version reads

$$\frac{\delta \Gamma_k[\Phi]}{\delta \phi}\Big|_{\phi=\phi_{\text{EoM}}(p)} = c_\phi(p), \tag{107}$$

where $\phi_{\text{EoM}}(p)$ has a non-trivial momentum-dependence for $R_s^\psi(p) \neq 0$. The computation of the respective part of the effective action requires the computation of momentum-dependent correlation functions which is a very challenging task. Instead, the fermion part of the flow implements an expansion about the momentum-dependent solution $\phi_{\text{EoM}}(p)$ in the shifted formulation. The solution $\phi_{\text{EoM}}$ is then built in as a background iteratively momentum shell by momentum shell.

Then, the (approximate) satisfaction of (106) can be checked for all $k$ and offers an additional self-consistency check. Moreover, if we keep the fermion loop in the flow equation, we still exploit the fact that the model is invariant under $\phi \to -\phi$, as induced by the chiral symmetry of the fermions. To be more specific, in the absence of any physical explicit symmetry breaking, the effective action should obey $\Gamma_k[\phi, \bar{\psi}, \psi] = \Gamma_k[-\phi, \bar{\psi}\exp(i\gamma_5), \exp(i\gamma_5)\psi]$. Assuming for convenience that we have integrated out the fermion such that we are left with a purely bosonic effective action $\Gamma_{\text{B},k}[\phi]$, the explicit symmetry breaking generated by the CS regulator in the flow can be restored by decomposing the effective action in an even (e) and odd (o) contribution in the field $\phi$,

$$\Gamma_{\text{B},k}[\Phi] = \Gamma_{\text{B},k}^{(e)}[\Phi] + \Gamma_{\text{B},k}^{(o)}[\Phi], \tag{108}$$

where

$$\Gamma_{\text{B},k}^{(e/o)}[\Phi] = \frac{1}{2}\big(\Gamma_{\text{B},k}[\Phi] \pm \Gamma_{\text{B},k}[-\Phi]\big). \tag{109}$$

In the limit $k \to 0$ and in the absence of any physical explicit chiral symmetry breaking (as associated with an explicit fermion mass term), the "physical effective action" is then given by the even part of the effective action. Those terms which violate the chiral symmetry are fully absorbed in the odd part. Note that the odd part is in general finite even in the limit $k \to 0$ as it contains all symmetry-breaking terms generated by the CS regulator in the RG flow. Moreover, the odd part is in general the dominant contribution for $k \to \Lambda$. In any case, the "physical effective action" in the IR limit is given by

$$\Gamma_{\mathrm{B},k=0}^{\mathrm{phys}}[\Phi] = \Gamma_{\mathrm{B},k=0}^{(e)}[\Phi] + c_{\phi}^{\mathrm{phys}}\phi \,. \tag{110}$$

Here, $c_{\mathrm{phys}}$ specifies the physical explicit symmetry breaking as, e.g., associated with an explicit fermion mass term. We close by noting that this decomposition can in principle also be implemented directly into the flow equation which provides us with a flow equation for the even part of the effective action which may be convenient in numerical studies.

A physically interesting example for such a Yukawa theory is the Quark-Meson (QM) model, often used for the study of the phase structure of QCD. In the QM model, one mostly uses quark regulators with a Dirac tensor structure. In [41] such a study is put forward in the local potential approximation (LPA) with CS regulators with a spatial UV cutoff, (30d), and four-dimensional momentum cutoff, (A.25). With these regulator choices the effective action satisfies (106). However, for momentum-dependent regulators, (106) is obtained with momentum-dependent shifts of the field. Hence, we expect that in LPA (106) can only be accommodated for $\Lambda \to \infty$.

### 5.2.5 Shift symmetry in vector Yukawa models and generalised Silver-Blaze property

The findings above extend to theories with vector bosons, as this allows to absorb fermionic regulators proportional to Dirac tensor structures. We briefly discuss this here on general grounds. Of course, this is interesting for applications in nuclear physics at high densities as well as for QCD close to a potential critical endpoint, where the density mode and the critical $\sigma$-mode mix. For illustrational purposes, we focus on the Hubbard-Stratonovich transformation of a four-fermi interaction in the vector channel. The respective vector modes $\omega_{\mu}$ are introduced similarly to the scalar–pseudo-scalar modes with

$$S_{\mathrm{yuk}}[\Phi] = \int_x \left\{ \bar{\psi} \left( \gamma_{\mu}\partial_{\mu} + m_{\psi} \right) \psi + h_{\omega}\,\bar{\psi}\,\gamma_{\mu}\omega_{\mu}\,\psi + \frac{m_{\omega}^2}{2}\omega_{\mu}^2 \right\}, \tag{111}$$

and the superfield

$$\Phi = (\psi, \bar{\psi}, \omega_{\mu}). \tag{112}$$

On the EoM of the vector field the second line in (111) corresponds to the following four-fermion interaction,

$$S_{\mathrm{4fermi}}[\psi, \bar{\psi}] = -\int_x \frac{h_{\omega}^2}{2m_{\omega}^2}\left( \bar{\psi}\,\gamma_{\mu}\,\psi \right)^2, \tag{113}$$

and hence the Yukakwa theory with the action (111) is identical with the purely fermionic theory with a four-fermi interaction in the vector channel.

Evidently, a Dirac regulator (79) can be absorbed in an imaginary shift of the vector field as follows

$$\gamma_{\mu}\omega_{\mu}(p) \to \gamma_{\mu}\omega_{\mu}(p) - \frac{1}{h_{\omega}}R_d^{\psi}(p), \qquad \epsilon R_d^{\psi}(p) \in \mathrm{i}\,\mathbb{R}. \tag{114}$$

We then have

$$S_{\text{yuk}} + \int_p \bar{\psi}(-p) R_d^{\psi}(p) \psi(p) \rightarrow S_{\text{yuk}} - \int c_{\omega,\mu} \omega_\mu \, , \tag{115}$$

where we have dropped field-independent terms on the right-hand side and

$$c_{\omega,\mu}(p) = \frac{m_\omega^2}{h_\omega} \frac{1}{4} \text{tr} \, \gamma_\mu R_d^{\psi}(p) \, . \tag{116}$$

However, this transformation introduces a complex-valued field $\omega_\mu$ and its integration contour can only be shifted back to the real axis in the absence of singularities. This is a generalisation of the Silver-Blaze property. Indeed, it is well-known that the chemical potential can be absorbed in $\omega_0$ (or the other way around) by considering $\bar{\mu}_\psi = \mu_\psi + h_\omega \omega_0$. In any case, we are eventually led to

$$\frac{\delta}{\delta R_d^{\psi}} \Big[ \Gamma_k[\Phi] + \int c_{\omega,\mu} \omega_\mu \Big] = 0 \, , \tag{117}$$

for $\omega_\mu$ smaller than the onset (pole position). As in the scalar case, we may still proceed with the fermionic regulator despite the identity (117). Then, the fermion part of the flow implements an expansion about the momentum-dependent solution $\omega_{\mu,\text{EoM}}(p)$ in the shifted formulation. The solution $\omega_{\mu,\text{EoM}}(p)$ is then built in as a background iteratively in the momentum-shell integrations.

## 5.3 Spectral renormalisation in gauge theories

The scope of the spectral Callan-Symanzik flow equations also extends to the particularly interesting case of gauge theories. Especially the non-perturbative infrared regime of QCD has been been studied intensively within the fRG approach [4, 64–72]. In this section we discuss the application of the spectral renormalisation group to gauge theories at the example of Yang-Mills theory, for respective works with the spectral DSE see [29, 63]. The classical gauge-fixed Yang-Mills action including the ghost term reads

$$S_{\text{YM}} = \int_x \Big[ \frac{1}{4} F_{\mu\nu}^a F_{\mu\nu}^a - \bar{c}^a \partial_\mu D_\mu^{ab} c^b + \frac{1}{2\xi} (\partial_\mu A_\mu^a)^2 \Big] \, . \tag{118}$$

Generally, setting up spectral flow equations for gauge theories works analogously as for scalar theories, discussed in section 5.1. The flow equations are derived in the usual manner, and spectral representations are used for the propagators of all fields, i.e. ghost and gluon propagator.

### 5.3.1 Ghost propagator

Formally, the ghost propagator is expected to obey the KL-representation [73, 74], if the corresponding propagator is causal. A recent direct calculation of the ghost spectral function with the spectral Dyson-Schwinger equation in [29] has confirmed this expectation. This computation has utilised a spectral representation for the gluon, which is discussed in section 5.3.2. Moreover, recent reconstructions [75, 76] show no signs of a violation of this property. It is found that the ghost spectral function exhibits a single particle peak at vanishing frequency with residue $1/Z_c$, whose value may depend on the non-perturbative infrared closure of the Landau gauge. Specifically, the scaling solution is obtained for the limit $Z_c \rightarrow 0$, see [29, 33]. In this case, the particle pole in the origin is no longer present. Instead, in the origin there

is the branch point of the non-integer power scaling law branch cut of the scaling solution. Note that in this case, the ordinary KL representation can no longer be applied, since the corresponding spectral function would show an IR divergence. For the current discussion, we will stick to the case of a massless particle pole in the IR.

Independent of the IR behaviour addition, a continuous scattering tail shows up in the spectral function via the logarithmic branch cut. This leads us to the general form of the ghost spectral function,

$$\rho_c(\omega) = \frac{\pi}{Z_c} \frac{\delta(\omega)}{\omega} + \tilde{\rho}_c(\omega), \tag{119}$$

where $\tilde{\rho}_c$ denotes the continuous tail of the spectral function. It has been shown in [29] that the ghost spectral function obeys an analogue of the Oehme-Zimmermann superconvergence property of the gluon [52, 53]. Expressed in terms of the spectral representation of the dressing, it reads

$$\int \frac{d\lambda}{\pi} \lambda \, \tilde{\rho}_c(\lambda) = -\frac{1}{Z_c} \,. \tag{120}$$

Equation (120) entails that the total spectral weight of the ghost vanishes. A generic discussion can be found in [29, 51].

Since the ghost spectral function (119) shows a (massless) particle pole, as for scalar theories, on-shell renormalisation conditions like (40) can be applied. This fixes the pole position of the scale-dependent ghost spectral function to $p^2 = -k^2$. In analogy to (119), the flowing ghost spectral function reads

$$\rho_{c,k}(\omega) = \frac{\pi}{Z_{c,k}} \frac{\delta(\omega - k) + \delta(\omega + k)}{\omega} + \tilde{\rho}_{c,k}(\omega), \tag{121}$$

where $\tilde{\rho}_{c,k}(\omega)$ has support for $|\omega| > 2k$. In the limit of vanishing cutoff, pole position and scattering onset move into the origin, and (119) is recovered.

### 5.3.2 Gluon propagator

The above discussion of the ghost spectral function and its existence was done under the assumption of a spectral representation of the gluon. In contrast to the ghost spectral function, there is an ongoing debate in the community whether or not this assumption is justified. In local QFTs only the existence of a spectral representation for asymptotic states is guaranteed. It has been argued that in Landau gauge this also applies to the gluon propagator [77–79]. While high precision spectral reconstructions are not in contradiction to this assumption and do work for the gluon propagator [33, 80–82], extensions with complex conjugate poles are also commonly entertained in reconstructions, see e.g. [75, 83–91]. A recent computation has shown, that the situation is indeed exceedingly intricate: its resolution may only be possible by also resolving the problem of a consistent non-perturbative gauge fixing [63]. The self-consistent implementation of the latter for propagators *and* vertices is subject to a non-perturbative infrared realisation of the respective Slavnov-Taylor identities. For a detailed discussion of the complex structure of Yang-Mills theory see [33, 63]. Specifically in [63] is has been shown that a solution of the Yang-Mills system with a spectral ghost and a non-spectral gluon would require non-trivial relations between the complex structures of vertices and propagators. In turn, while less conclusive, in [63] we have also found numerical indications, that a self-consistent solution system with spectral representations for both ghost and gluon propagators, if existent, may also require self-consistent or rather STI-consistent solutions for non-trivial vertices.

In the present work we add nothing new to the resolution of this intricate problem, but simply consider the flow of the gluon spectral function under the assumption of its existence. Likewise, we assume a spectral representation for the ghost, with a pole at $\omega^2 = k^2$, c.f. (121). The branch point of the ghost loop contribution to the gluon propagator's branch cut lies at $\omega^2 = (2k)^2$. Due to the massless nature of the ghost, the position of the branch point in the gluon propagator thus necessarily is in the origin for vanishing cutoff scale, $k = 0$. However, due to the lack of a gluon particle peak, a direct identification of a flowing mass scale $k$ as in the scalar theory section 5.1, is not possible for the gluon. Consequently, there is no unique way to stop the flow at some $k_{\text{IR}} = m_{\text{phys}}$, where the physical limit of the theory is recovered. Furthermore, the lack of unique gluon mass scale entails that we cannot use on-shell renormalisation here. Eventually, we wish to recover the IR behaviour of the gluon propagator known from other non-perturbative studies, e.g. via functional approaches [69, 92, 93]. In consequence, we can define the IR scale only implicitly, and $k_{\text{IR}}$ depends directly on the initial conditions employed. This poses the question of how to consistently couple the gluonic flow to that of the ghost. A consistent, coupled flow is required to simultaneously reach the explicitly resp. implicitly defined IR scales $k_{\text{IR}}^{(\text{ghost})} = 0$ and $k_{\text{IR}}^{(\text{gluon})}$. This can be implemented by flowing both equations with a common scale $k$ down to 0, where the IR limit of the ghost propagator is reached. We then proceed to further lower $k$ solely in the gluon propagator flow equation down to the point where we reach $k_{\text{IR}}^{(\text{gluon})}$ defined by, e.g. scaling as IR behaviour, c.f. [69, 93]. Note that this procedure needs to be supplemented with an appropriate choice of initial conditions guaranteeing $k_{\text{IR}}^{(\text{gluon})} \leq 0$. This clarifies that the described procedure of flowing with two seemingly different scales simply amounts to an implicit choice of initial conditions and does not lead to an inconsistency between the different flow equations. In such a procedure, adjusting the initial conditions is similar to common fRG calculations. We therefore expect a similar fine-tuning problem for the Yang-Mills system as for example encountered in [69].

The proper choice of initial conditions comes in case of the gluon propagator with another technical complication. It is well-known that in massive Yang-Mills theory, the gluon propagator exhibits complex-conjugate poles. It has been demonstrated in [63] that these can also violate the spectral representation of the ghost propagator, in turn inducing a cascade of non-analyticities in both propagators. Since the Callan-Symanzik cutoff effectively constitutes a mass term, the construction of an initial condition respecting the spectral representation poses a crucial challenge. On the other hand, using modified spectral representations that explicitly take into account complex singularities [63], one is able to track the evolution of the complex poles through the flow. This allows to make a statement about their existence in the full correlation function at $k_{\text{IR}}$. It has been studied e.g. in [25] how regulator-induced poles vanish in the $k \to 0$ limit in a quantum mechanical system.

## 5.4 Spectral renormalisation in asymptotically safe gravity

The present approach including the use of the spectral BPHZ-renormalisation has already been applied in [24] to asymptotically safe gravity. The respective classical action is the Einstein Hilbert (EH) action,

$$S_{\text{EH}}[g_{\mu\nu}] = \frac{1}{16\pi G_{\text{N}}} \int \mathrm{d}^4 x \, |\det g_{\mu\nu}|^{\frac{1}{2}} \left( \mathcal{R} - 2\Lambda \right), \qquad (122)$$

with Newton's coupling $G_{\text{N}}$, the curvature scalar $\mathcal{R}$ and the cosmological constant $\Lambda$. The dynamical quantum field is the metric $g_{\mu\nu}$. The EH action is then augmented with a gauge-

fixing term given by

$$S_{\mathrm{gf}}[\bar{g}, h] = \frac{1}{2\alpha} \int \mathrm{d}^4 x \sqrt{\bar{g}} \, \bar{g}^{\mu\nu} F_\mu F_\nu, \tag{123}$$

with the gauge-fixing condition $F_\mu$

$$F_\mu[\bar{g}, h] = \bar{\nabla}^\nu h_{\mu\nu} - \frac{1+\beta}{4} \bar{\nabla}_\mu h^\nu{}_\nu. \tag{124}$$

The respective ghost action reads

$$S_{\mathrm{gh}}[\bar{g}, \phi] = \int \mathrm{d}^4 x \sqrt{\bar{g}} \, \bar{c}^\mu M_{\mu\nu} c^\nu, \tag{125}$$

with the Faddeev-Popov operator

$$M_{\mu\nu} = \bar{\nabla}^\rho \big( g_{\mu\nu} \nabla_\rho + g_{\rho\nu} \nabla_\mu \big) - \frac{1+\beta}{2} \bar{g}^{\sigma\rho} \bar{\nabla}_\mu g_{\nu\sigma} \nabla_\rho. \tag{126}$$

The gauge-fixing sector enforces the introduction of a background metric $\bar{g}_{\mu\nu}$ as the full metric would introduce unwanted interaction terms to the gauge fixing. For the discussion of spectral flows we use the flat Minkowski metric as a background, $\bar{g}_{\mu\nu} = \eta_{\mu\nu}$. Amongst other reasons this choice is taken as spectral representations in the presence of non-trivial backgrounds pose additional conceptual intricacies. Furthermore, we use a linear split of the full metric,

$$g_{\mu\nu} = \eta_{\mu\nu} + \sqrt{16\pi G_{\mathrm{N}}} \, h_{\mu\nu}, \tag{127}$$

and the fluctuation field $h_{\mu\nu}$ carries the full dynamics of quantum gravity. For more details on this fluctuation approach to gravity see [94].

In [24], the spectral flow of the graviton propagator has been computed with the spectral CS equation. First of all, this has provided a non-trivial existence proof of the graviton spectral representation within the approximation discussed there. This is specifically remarkable, given the ongoing discussion concerning the existence of a spectral representation for the gluon, see section 5.3. Furthermore, in [24] explicit numerical results for the spectral function $\rho_h$ have been obtained: the spectral function is positive but not normalisable due to the large positive UV anomalous dimension $\eta_h \approx 1$, see the discussion in section 4.2.

Here, we show that the momentum structure of asymptotically safe propagators and vertices allows for a renormalised spectral CS flows with a finite number of counter terms: to begin with, the loops in quantum gravity have the same spectral representation displayed in (58). As in four-fermi models in four space-time dimensions, the theory is perturbatively non-renormalisable. Moreover, already classical vertices involve general powers of the graviton, $S_{\mathrm{EH}}^{(n)} \neq 0$ for all $n \in \mathbb{N}$, and further ones are generated by loop corrections.

Since the seminal Euclidean fRG paper of Reuter [95] quite some further evidence has been accumulated for quantum gravity being asymptotically safe [96, 97], for recent fRG-reviews see e.g. [4, 94, 98]. This scenario is based on a non-trivial ultraviolet fixed point, the Reuter fixed point. In the fRG setting it implies

$$\lim_{k \to \infty} G_{\mathrm{N},k} k^2 = g_{\mathrm{N}}^*, \qquad G_{\mathrm{N},k=0}(p \to \infty) \to \frac{g_{\mathrm{N}}^*}{p^2}, \tag{128}$$

where $g_{\mathrm{N}}^*$ is the fixed point of the dimensionless Newton's coupling, and $G_{\mathrm{N},k} = G_{\mathrm{N},k}(p = 0)$. Typically, for fixed point investigations the (unphysical) limit $k \to \infty$ with $p \ll k$ is used, as this limit is technically less challenging and the $k$-scaling and fixed point 'couplings' reflect the physical momentum scaling and fixed point coupling.

For our discussion of the spectral setting it is important to note that the asymptotically safe Newton's coupling *necessarily* decays with $1/p^2$ for large momenta with the respective FP equation,

$$\lim_{p^2 \to \infty} \frac{\partial_{p^2} G_{\mathrm{N},k=0}(p)}{p^2} = 0 \,, \tag{129}$$

which is reflected in $\lim_{k\to\infty} \partial_t g_{\mathrm{N}} = 0$ with $g_{\mathrm{N}} = k^2 G_{\mathrm{N},k}$. These considerations entail that a convenient parametrisation of $h_{\mu\nu}$-vertices is given by

$$\Gamma_{h^n}^{(n)}(p_1,...,p_n) = \prod_{i=1}^{n} Z_h^{1/2}(p_i) \bar{\Gamma}_{h^n}^{(n)}(p_1,...,p_n) \,, \tag{130}$$

where the $Z_h^{1/2}$ factors take care of the RG-running of the legs and $\bar{\Gamma}_{h^n}^{(n)}$ shows the momentum running of a (vertex) coupling. Accordingly, these vertex dressings decays with powers of $\bar{p}$ at a symmetric point with $p_i^2 = \bar{p}^2$: in terms of vertex avatars of Newton's coupling, $G_{\mathrm{N},n}$, the symmetric point dressing $\bar{\Gamma}_{h^n}^{(n)}(\bar{p})$ is proportional to $G_{\mathrm{N},n}^{n/2-1}(\bar{p})$. In the asymptotically safe UV regime all these couplings have to decay with $1/\bar{p}^2$ and we are led to

$$\lim_{\bar{p}\to\infty} \bar{\Gamma}_{h^n}^{(n)}(\bar{p}) \propto \lim_{\bar{p}\to\infty} \bar{p}^2 G_{\mathrm{N},n}^{n/2-1}(\bar{p}) \propto \bar{p}^2 \left(\frac{1}{\bar{p}^2}\right)^{\frac{n}{2}-1} \,. \tag{131}$$

Inserting this scaling back in the loop equations shows the consistency of this scaling: the UV momentum scaling of all diagrams is given by

$$\lim_{\bar{p}\to\infty} \partial_t \bar{\Gamma}_{h^n}^{(n)}(\bar{p}) \propto \bar{p}^2 \left(\frac{1}{\bar{p}^2}\right)^{\frac{n}{2}} \,, \tag{132}$$

which is exactly that of $\bar{\Gamma}^{(n)}/\bar{p}^2$. In standard perturbation theory, the running would be $\partial_t \bar{\Gamma}^{(n)} \propto \bar{p}^2 \bar{\Gamma}^{(n)}$ related to the perturbative non-renormalisability of the theory. Here, one additional $1/\bar{p}^2$ scaling stems from the second propagator in the cutoff line $G_k \partial_t R_k G_k$ and reflects the reduction of the UV degree of divergence by two in the CS flow in comparison to standard loop diagrams. The other $1/\bar{p}^2$ scaling stems from the fixed-point scaling of Newton's coupling, which effectively shifts the theory to its critical dimension.

In summary we deduce that the only diagrams that require renormalisation via $\partial_t S_{\mathrm{ct}}$ are that of $\bar{\Gamma}^{(2)}$. In turn, the flows of $\bar{\Gamma}^{(n>2)}$ are finite but the renormalisation conditions in gravity for $\mu \to 0$ should lead to the Einstein-Hilbert action, which uniquely fixes the full $\partial_t S_{\mathrm{ct}}$ in the CS flow (39). In summary, a fully consistent spectral CS flow requires also the inclusion of momentum-dependent vertex functions. However, the above analysis also entails that constant vertex approximations can be entertained. In this case the *finite* subtractions $\partial_t S_{\mathrm{ct}}^{(n)}$ are elevated to the standard subtraction of counter terms with the constraint of leaving the IR limit of the Einstein-Hilbert action intact. In any case it leaves us with a finite number of relevant couplings given by those obtained with a spectral spatial momentum regulator.

## 6  Conclusions

We close this work with a brief summary of the main results, detailed discussions can be found in the respective Sections.

In the present work we have derived a novel functional flow equation with *flowing* renormalisation, see (38) in section 3.2. Flowing renormalisation entails that the renormalisation

condition can be adapted with the flowing scale. This can be used for fRG flows from the finite renormalised UV effective action at large infrared cutoff scales to the full effective action at $k = 0$. Importantly, it also allows for manifestly finite fRG flows with regulators that do not implement a UV regularisation of the loop, such as the Callan-Symanzik regulator. The respective CS flow, (39), is manifestly finite in general truncation schemes. The novel fRG flows (38) and (39) constitute key results of the present work.

While finite (or homogeneous) CS equations are well-known, they are based on multiplicative renormalisation, which is not amiable to general truncation schemes, and in particular do not support most non-perturbative schemes. In turn, the present derivation is solely based on the general fRG framework with finite flow equations with respect to an infrared regulator. This embeds the Callan-Symanzik equation self-consistently in this Wilsonian framework. The current derivation also provides the full formal justification of its use in asymptotic safety [24].

Notably, the current derivation does not require coupling redefinitions at each RG-step proportional to $\partial_{\lambda_\phi} \Gamma_k$. These terms can be added by augmenting the current flow with a full homogeneous RG transformation, see [32] which reduces (39) to a more standard form of the CS flow. The computation of such terms is feasible but constitutes a considerable additional technical challenge, for respective considerations in a different context see [50].

We have then used the finite Lorentz invariant CS flows to set-up the Lorentz invariant *spectral fRG* in section 4. We have then discussed the spectral fRG in scalar theories, Yukawa theories, gauge theories and quantum gravity in section 5. In short, the spectral fRG is a simple finite, ready to use, spectral form of the Callan-Symanzik equation, and we hope to report on respective results in the above theories in the near future.

# Acknowledgments

We thank Jannik Fehre, Markus Huber, Daniel Litim, Johannes Roth, Bernd-Jochen Schaefer and Lorenz von Smekal for discussions and collaborations on related projects.

**Funding information** This work is done within the fQCD collaboration [99]. It is funded by the Deutsche Forschungsgemeinschaft (DFG, German Research Foundation) under Germany's Excellence Strategy EXC 2181/1 - 390900948 (the Heidelberg STRUCTURES Excellence Cluster), the Collaborative Research Centre SFB 1225 (ISOQUANT), and EMMI. JB acknowledges support by the DFG under grant BR 4005/6-1 (Heisenberg program). Moreover, JB and AG acknowledge support by the Deutsche Forschungsgemeinschaft (DFG, German 1355 Research Foundation) – Projektnummer 279384907 – SFB 1245. JB and NW acknowledge support by the Deutsche Forschungsgemeinschaft (DFG, German Research Foundation) – Project number 315477589 – TRR 211. JB and NW acknowledge support by the State of Hesse within the Research Cluster ELEMENTS (Project No. 500/10.006). WJF is supported by the National Natural Science Foundation of China under Grant No. 12175030. MR acknowledges support by the Science and Technology Research Council (STFC) under the Consolidated Grant ST/T00102X/1. JB and ST acknowledge support by BMBF under grant 05P20RDFCA. JH und FI acknowledge support by the Studienstiftung des Deutschen Volkes.

# A Renormalised CS flow of the effective potential

We consider the 0th order derivative expansion (LPA) of a $\phi^4$-theory in four dimensions with the classical action

$$S_\phi[\phi] = \int_x \left[ \frac{1}{2} \phi \left( -\partial^2 + m_\phi^2 \right) \phi + \frac{\lambda_\phi}{4!} \phi^4 \right], \tag{A.1}$$

see also (3) and (63). The effective action in LPA is given by

$$\Gamma_{k,\Lambda}[\phi] = \int_x \left\{ \frac{1}{2} (\partial_\mu \phi)^2 + V_{k,\Lambda}(\phi) \right\}, \tag{A.2}$$

and the flow equation for the effective potential reads

$$\left( \partial_t + \mathcal{D}_k \partial_{t_\Lambda} \right) V_{k,\Lambda}(\phi) = \frac{1}{2} \int_p \frac{\left( \partial_t|_\Lambda + \mathcal{D}_k \partial_{t_\Lambda} \right) R_{k,\Lambda}^\phi(p)}{p^2 + V_{k,\Lambda}^{(2)}(\phi) + R_{k,\Lambda}^\phi(p)} . \tag{A.3}$$

In the following, we explicitly derive the counter term action in (A.3) and discuss the finiteness of the CS flow as well as the flowing renormalisation. We remark that already from (A.3) we deduce that the counter term action flow is a function of $V^{(2)}(\phi)$: the $\Lambda$-part flow is peaked at $p^2 \approx \Lambda^2$ and hence in the limit $\Lambda \to \infty$ it necessarily depends on the dimensionless ratio $(V_k^{(2)} + k^2)/\Lambda^2$. Together with the prefactor $\mathcal{D}_k$ and the requirement of $\Lambda$-independence and finiteness of the flow we deduce

$$\partial_t S_{\text{ct}}(\phi) = \sum_{i=1}^{N_{\max}} c_i \int_x \left( \frac{k^2 + V_k^{(2)}(\phi)}{\Lambda_{m_\phi}^2} \right)^i, \tag{A.4}$$

where $N_{\max}$ is the number of UV relevant coupling parameters in the effective potential, and $\Lambda_{m_\phi}$ is linked to the mass renormalisation of the theory. For example, we can take the physical mass scale of the $\phi^4$-theory. The coefficients $c_i$ are adjusted such that they render the full flow $\Lambda$-independent. For standard IR regulators with the properties (7) the $c_i$ are manifestly $\Lambda$-independent as the IR flow is. In turn, for the CS regulator, the IR flow part in (A.3) depends on the UV cutoff $\Lambda$ and the subtraction with the counter term action flow (A.4) renders the left hand side of (A.3) $\Lambda$-independent and finite. In the following we will derive $\partial_t S_{\text{ct}}$ with different UV regularisations and compute (A.4) explicitly.

For a comparison to the standard UV renormalisation as implemented in perturbation theory we also parametrise the effective potential in a Taylor series,

$$V_k(\phi) = \sum_{n=1}^\infty \frac{\lambda_{2n,k}}{2n!} \phi^{2n}, \tag{A.5}$$

where we have dropped the subscript $_\Lambda$ for the sake of readability. In four dimensions, the potential only hosts a relevant coupling $m_{\phi,k}^2 = \lambda_{2,k}$ and one marginal coupling, $\lambda_{\phi,k} = \lambda_{4,k}$. A respective flow of the counter term action in perturbation theory is then given by

$$\partial_t S_{\text{ct}}[\phi] = \frac{1}{2} \partial_t \delta m_{\phi,k}^2 \phi^2 + \frac{1}{4!} \partial_t \delta \lambda_{\phi,k} \phi^4, \tag{A.6}$$

with the two (flowing) renormalisation parameters $\partial_t \delta m_{\phi,k}^2$ and $\partial_t \delta \lambda_{\phi,k}$. In the following we derive the non-perturbative analogue of (A.6) in LPA.

### A.1 CS flow with dimensional regularisation

First of all, we can use a bootstrap approach and assume that the CS limit, i.e. $R_{k,\Lambda} \to k^2$, exists if the UV-regularisation is chosen appropriately. We also remark that the degree of divergence is lowered by two if considering $\partial_t V_k^{(1)}(\phi)$. Using dimensional regularisation with $d = 4 - 2\epsilon$ in the loop integral in (A.3), we arrive at

$$\partial_t V_k^{(1)}(\phi) = -\mu^{2\epsilon} \int \frac{d^d p}{(2\pi)^d} \frac{k^2 V_k^{(3)}(\phi)}{\left[p^2 + k^2 + V_k^{(2)}(\phi)\right]^2} - \partial_t S_{\text{ct}}^{(1)}(\phi), \tag{A.7}$$

where we have used that with dimensional regularisation both the counter term flow and loop integral in the flow equation are separately finite. Equation (A.7) is readily integrated, leading to

$$\partial_t V_k^{(1)}(\phi) = -\frac{k^{2-2\epsilon} \mu^{2\epsilon}}{(4\pi)^{(2-\epsilon)}} \Gamma(\epsilon) V_k^{(3)}(\phi) \left[1 + \frac{V_k^{(2)}(\phi)}{k^2}\right]^{-\epsilon} - \partial_t S_{\text{ct}}^{(1)}(\phi), \tag{A.8}$$

which includes a divergent term due to $\Gamma(\epsilon) = 1/\epsilon - \gamma + O(\epsilon)$ with the Euler-Mascheroni constant $\gamma \approx 0.577$. Thus, finiteness of (A.8) basically dictates a counter term flow,

$$\partial_t S_{\text{ct}}^{(1)} = -\frac{k^2}{(4\pi)^2} \left(\frac{1}{\epsilon} - \gamma - \log \frac{\Lambda_{m_\phi}^2}{4\pi\mu^2}\right) V_k^{(3)}(\phi). \tag{A.9}$$

The flow of the counter term in (A.9) is proportional to the third derivative of the full effective potential, $V^{(3)}(\phi)$. This is dictated by the necessity of cancelling the $1/\epsilon$-term in the loop integral, and simply is the required mass renormalisation of the four-dimensional $\phi^4$-theory as in (A.6). The $\Lambda_{m_\phi}$ carries the respective renormalisation condition. However, in LPA all quantities are $\phi$-dependent, and (A.9) reflects the field dependence of the mass $V^{(2)}(\phi)$.

Inserting the counter term flow (A.9) in (A.8) leads us to the final renormalised CS flow,

$$\partial_t V_k^{(1)}(\phi) = \frac{k^2}{(4\pi)^2} V_k^{(3)}(\phi) \log \frac{k^2 + V_k^{(2)}(\phi)}{\Lambda_{m_\phi}^2}. \tag{A.10}$$

In (A.10) the dependence on the renormalisation scale $\mu$ has been traded for one on the renormalisation condition that carries the value of the physical mass. We close this analysis with the remark that for $d < 4$ the flows $\partial_t V_k^{(1)}(\phi)$ are finite in dimensional regularisation. For $d = 2$ we have a convergent integral, while in odd dimensions dimensional regularisation provides finite results in the first place, implying a renormalisation by construction. While then the necessity of a counter term flow is absent, a flowing renormalisation requires it. This is discussed below and in the following Sections (see in particular appendices A.4 and A.5), where momentum regulators are used.

Equation (A.9) is the counter term necessary to render the flow finite. However, as discussed above, we can also use the present setup to implement further flowing renormalisation conditions also for couplings that do not require regularisation. Within the present example, a general counter term can be written as

$$\partial_t S_{\text{ct}}^{(1)} = -\frac{k^2}{(4\pi)^2} \left[\frac{1}{\epsilon} - \gamma - \log \frac{\Lambda_{m_\phi}^2}{4\pi\mu^2} - \sum_{i=2}^{N_{\max}} c_i \left(\frac{k^2 + V_k^{(2)}(\phi)}{\Lambda_{m_\phi}^2}\right)^i\right] V_k^{(3)}(\phi). \tag{A.11}$$

The last terms $\sim c_i$ follow from (A.4). As we will demonstrate now, these additional terms renormalise higher order couplings, and the coefficients $c_i$ are uniquely fixed by the flowing renormalisation conditions discussed in section 3.3.

For the sake of simplicity, we consider a $\phi^4$ theory in the symmetric phase, i.e. $\bar{\phi} = 0$. Similarly to (41), we impose the following renormalisation conditions at $\mu = 0$,

$$\partial_t V_k^{(2)}(\bar{\phi}) = \partial_t m_{\phi,k}^2 = 0\,,$$
$$\partial_t V_k^{(4)}(\bar{\phi}) = \partial_t \lambda_{\phi,k} = 0\,. \tag{A.12}$$

Note that in LPA the wave function renormalisation is always $Z = 1$, so renormalisation is unnecessary. In contrast to the on-shell conditions in (41), these conditions at $\mu = 0$ enforce that the quadratic and quartic couplings, $m_{\phi,k}^2$ and $\lambda_{\phi,k}$, do not run. This can be achieved by taking $N_{\max} = 2$ in (A.11). The flow of $V_k^{(1)}$ is then

$$\partial_t V_k^{(1)}(\phi) = \frac{k^2}{(4\pi)^2} \left( \log \frac{k^2 + V_k^{(2)}(\phi)}{\Lambda_{m_\phi}^2} - c_{\lambda_\phi} \frac{k^2 + V_k^{(2)}(\phi)}{\Lambda_{m_\phi}^2} \right) V_k^{(3)}(\phi)\,. \tag{A.13}$$

By taking functional derivatives and setting $\phi = \bar{\phi}$ in the end, we find for the quadratic coupling

$$\partial_t m_{\phi,k}^2 = \frac{k^2}{(4\pi)^2} \left( \log \frac{k^2 + m_{\phi,k}^2}{\Lambda_{m_\phi}^2} - c_{\lambda_\phi} \frac{k^2 + m_{\phi,k}^2}{\Lambda_{m_\phi}^2} \right) \lambda_{\phi,k}\,, \tag{A.14}$$

which corresponds to a tadpole diagram. For the quartic coupling we find

$$\partial_t \lambda_{\phi,k} = \frac{k^2}{(4\pi)^2} \left[ \left( \log \frac{k^2 + m_{\phi,k}^2}{\Lambda_{m_\phi}^2} - c_{\lambda_\phi} \frac{k^2 + m_{\phi,k}^2}{\Lambda_{m_\phi}^2} \right) \lambda_{6,k} + 3 \left( \frac{1}{k^2 + m_{\phi,k}^2} - \frac{c_{\lambda_\phi}}{\Lambda_{m_\phi}^2} \right) \lambda_{\phi,k}^2 \right]\,, \tag{A.15}$$

which corresponds to a tadpole and a fish diagram. Through the renormalisation conditions (A.12), the two parameters $\Lambda_{m_\phi}^2$ and $c_{\lambda_\phi}$ are uniquely fixed to be

$$\Lambda_{m_\phi}^2 = \frac{k^2 + m_\phi^2}{e}\,, \qquad c_{\lambda_\phi} = \frac{1}{e}\,, \tag{A.16}$$

where we used $m_\phi^2 = m_{\phi,k}^2$, as it does not run. In general, in theories with fundamental higher-order couplings $\lambda_{2i>4,k}$, the respective renormalisation constants $c_{i>2}$ are required. To be more precise, each $c_i$ can be used to renormalise a different type of diagrammatic contribution to the flow, namely the one containing $i$ vertices.

## A.2 Infrared CS regulator and a sharp ultraviolet cutoff

We proceed with an analysis of the counter term triggered by a sharp UV cutoff together with a CS infrared cutoff with

$$R_{k,\Lambda}(p) = k^2 \frac{1}{\theta(\Lambda^2 - p^2)}\,, \tag{A.17}$$

see also (30e). As discussed there, such a choice also introduces a UV regularisation of the theory, and not only one for the flow itself as the standard IR regulators. The scalar CS flows in [38] have been computed with this choice, which leads to flows with a *physical* UV cutoff as discussed there.

Note that the highest order divergence in the flow is the one of the field-independent term. In perturbation theory it is of the order $\Lambda^d$ for a $d$-dimensional theory. For the CS flow this divergence is reduced to $\Lambda^{d-2}$ as also discussed in appendix A.1. It is related to the unphysical vacuum energy and we eliminate it by taking a $\phi$-derivative of the flow which leads us to

$$\left(\partial_t + \mathcal{D}_k \partial_{t_\Lambda}\right) V_k^{(1)}(\phi) = -\frac{1}{2} V_k^{(3)}(\phi) \int \frac{d^d p}{(2\pi)^d} \frac{\left(\partial_t\big|_\Lambda + \mathcal{D}_k \partial_{t_\Lambda}\right) R_{k,\Lambda}^\phi(p)}{\left[p^2 + V_k^{(2)}(\phi) + R_{k,\Lambda}^\phi(p)\right]^2}, \tag{A.18}$$

where we have dropped the subscript $_\Lambda$ in the effective potential. With the regulator (A.17) the flow (A.18) turns into

$$\partial_t V_k^{(1)}(\phi) = -\int_{p^2 \le \Lambda^2} \frac{d^d p}{(2\pi)^d} \frac{k^2 V_k^{(3)}(\phi)}{\left[p^2 + k^2 + V_k^{(2)}(\phi)\right]^2} + \mathcal{D}_k \frac{\Lambda^4}{(4\pi)^2} \frac{V_k^{(3)}(\phi)}{\Lambda^2 + k^2 + V_k^{(2)}(\phi)}, \tag{A.19}$$

where we have used that with the regulator (A.17) we have

$$\begin{aligned}
G_{k,\Lambda}\left(\partial_{t_\Lambda} R_{k,\Lambda}\right) G_{k,\Lambda} &= -\partial_{t_\Lambda} G_{k,\Lambda} - G_{k,\Lambda}\left(\partial_{t_\Lambda} \Gamma_k^{(2)}\right) G_{k,\Lambda} \\
&= -\partial_{t_\Lambda}\left(\frac{1}{k^2 + \Gamma_k^{(2)}} \theta(\Lambda^2 - p^2)\right) - G_{k,\Lambda}\left(\partial_{t_\Lambda} \Gamma_k^{(2)}\right) G_{k,\Lambda} \\
&= -\frac{1}{p^2 + k^2 + V_k^{(2)}(\phi)} \partial_{t_\Lambda} \theta(\Lambda^2 - p^2) \\
&= \frac{2\Lambda^2 \delta(p^2 - \Lambda^2)}{\Lambda^2 + k^2 + V_k^{(2)}(\phi)}.
\end{aligned} \tag{A.20}$$

From the second to the third line in (A.20) we have used that we have a sharp UV cutoff. The $\partial_{t_\Lambda} \Gamma^{(2)}$-terms ( in LPA $\partial_{t_\Lambda} V^{(2)}$-terms) from the first and second term in the third line cancel each other, which leads us to the fourth line and the final result.

The first line in (A.19) is also integrated easily and we arrive at

$$\partial_t V_k^{(1)}(\phi) = \frac{k^2}{(4\pi)^2} V_k^{(3)}(\phi) \left[1 + \log\left(\frac{k^2 + V_k^{(2)}(\phi)}{\Lambda^2}\right) + \frac{\Lambda^2}{k^2} \mathcal{D}_k \frac{1}{1 + \frac{k^2 + V_k^{(2)}(\phi)}{\Lambda^2}}\right], \tag{A.21}$$

where we have already assumed the limit $\Lambda \to \infty$ and dropped some subleading terms. The first line in (A.21) matches that in (A.10) obtained from dimensional regularisation as required.

The necessity of cancelling the divergent term from the IR flow (in the CS limit) proportional to $V_k^{(3)}(\phi) \log \Lambda^2$ leads to

$$\mathcal{D}_k = -\frac{k^2}{\Lambda^2} \log \frac{\Lambda_{m_\phi}^2}{\Lambda^2}, \tag{A.22}$$

where $\Lambda_{m_\phi}^2$ carries the renormalisation condition of the mass. The choice (A.22) suffices to render the flow of the potential finite. The flow of the counter term provided by (A.22) is uniquely given by

$$\partial_t S_{\text{ct}}^{(1)}[\phi] = \left[\frac{k^2}{(4\pi)^2} \log \frac{\Lambda_{m_\phi}^2}{\Lambda^2}\right] V^{(3)}(\phi). \tag{A.23}$$

We find $\mathcal{D}_k > 0$ for $\Lambda_{m_\phi} < \Lambda$, as well as $\mathcal{D}_k \to 0$ for $\Lambda \to \infty$.

We emphasise that (A.23) only depends on one parameter, its field-dependence is uniquely fixed. Equation (A.23) simply encodes the mass renormalisation of the $\phi^4$ theory, and the non-polynomial field dependence solely originates in the approximation used. It agrees with the counter term flow in dimensional regularisation, (A.9), and is the LPA version of the perturbative counter term flow (A.6).

With (A.22), the flow (A.21) reads in the limit $\Lambda \to \infty$

$$\partial_t V_k^{(1)}(\phi) = \frac{k^2}{(4\pi)^2} V_k^{(3)}(\phi) \left[ 1 + \log\left( \frac{k^2 + V_k^{(2)}(\phi)}{\Lambda_{m_\phi}^2} \right) \right], \tag{A.24}$$

which is our final *renormalised* result for the Callan-Symanzik flow of the effective potential. Naturally, it agrees with the renormalised flow obtained with dimensional regularisation (A.10), as all dependence on the UV regularisation is removed with the renormalisation.

As already discussed before, (A.24) depends on one parameter, $\Lambda_{m_\phi}$, that allow us to adjust the flowing renormalisation conditions for the mass $m_{\phi,k}^2$, or, alternatively for the coupling $\lambda_{\phi,k}$. However, the $\phi^4$-theory in four dimension has one relevant, $m_\phi^2$, and two marginal, $\lambda_\phi$ and $Z_\phi$, parameters. In the current approximation $Z_\phi = 1$ and does not require renormalisation. Hence we are left with two parameters, $m_\phi^2, \lambda_\phi$, whose flow or lack thereof at the renormalisation point $p = \mu$ can be adjusted by $\Lambda_{m_\phi}$ and a further parameter $c_{\lambda_\phi}$. This parameter is not present, if we only consider the regulator (A.17). The second (and third) parameter can only be introduced by also changing the regulator itself and not only the scale, see also the discussion in section 3.2.

## A.3 Sharp ultraviolet regularisation of the CS regulator

The change of the regulator shape function can be introduced by simply considering a combination of two different regulators and varying their linear combination during the flow. For that purpose we first discuss another natural choice for a regulator that also introduces a UV cutoff in the flow. We consider a CS mass term that vanishes for large momenta,

$$R_{k,\Lambda}(p) = k^2 \theta\left( \Lambda^2 - p^2 \right), \tag{A.25}$$

see also (30d). This leaves the CS part of the flow in (A.19) unchanged, but the UV flow changes. As discussed below (30d), this only regularised the flow itself but does not provide a UV cutoff for the theory. We use

$$G_{k,\Lambda} \partial_{t_\Lambda} R_{k,\Lambda} G_{k,\Lambda} = - \left. \partial_{t_\Lambda} \right|_{V_k^{(2)}} \left[ \frac{\theta(\Lambda^2 - p^2)}{p^2 + k^2 + V_k^{(2)}(\phi)} + \frac{\theta(p^2 - \Lambda^2)}{p^2 + V_k^{(2)}(\phi)} \right]$$

$$= 2\Lambda^2 \delta(p^2 - \Lambda^2) \left[ \frac{1}{\Lambda^2 + k^2 + V_k^{(2)}(\phi)} - \frac{1}{\Lambda^2 + V_k^{(2)}(\phi)} \right]$$

$$= -\frac{2k^2}{\Lambda^2} \frac{\delta(p^2 - \Lambda^2)}{\left( 1 + \frac{k^2 + V_k^{(2)}(\phi)}{\Lambda^2} \right) \left( 1 + \frac{V_k^{(2)}(\phi)}{\Lambda^2} \right)}. \tag{A.26}$$

Note that (A.26) is suppressed with $1/\Lambda^2$ in comparison to (A.20). This originates in the fact that the regulator (A.25) only is an infrared regulator and the $\Lambda$-flow encodes the change of

the UV cutoff in the flow, whose UV divergence is lowered by two in comparison to diagrams e.g. in DSEs. In turn, (A.17) leads to a UV regularisation of the theory and the $\Lambda$-flow encodes the change of the UV cutoff in the theory. This difference explains both, the relative $1/\Lambda^2$ factor and the relative minus sign.

Using (A.26) in the flow equation (A.18) we are led to

$$\partial_t V_k^{(1)}(\phi) = \frac{k^2}{(4\pi)^2} V_k^{(3)}(\phi) \left[ 1 + \log\left(\frac{k^2 + V_k^{(2)}(\phi)}{\Lambda^2}\right) - \mathcal{D}_k \frac{1}{\left(1 + \frac{k^2 + V_k^{(2)}(\phi)}{\Lambda^2}\right)\left(1 + \frac{V_k^{(2)}(\phi)}{\Lambda^2}\right)} \right]. \tag{A.27}$$

Taking the limit $\Lambda \to \infty$ in (A.27) with the choice

$$\mathcal{D}_k = \log\frac{\Lambda_{m_\phi}^2}{\Lambda^2}, \tag{A.28}$$

we are led to the same flow of the counter term (A.23) as in appendix A.2. In summary, in both cases as well as in dimensional regularisation the flow of the counter term is fixed uniquely, and we are led to the same renormalised flow, (A.10) or (A.24).

## A.4 CS flow with flowing renormalisation

In both these cases we arrive at finite flows, but cannot use the full power of flowing renormalisation. To that end we combine the two examples in one regulator. We define

$$R_{k,\Lambda} = k^2\left(\theta\left(\Lambda^2 - p^2\right) + \left[\frac{1}{\theta\left(\Lambda_{\mathrm{UV}}^2 - p^2\right)} - 1\right]\right). \tag{A.29}$$

The regulator in (A.29) is a combination of the CS-type infrared regulator (A.25) with a UV cutoff $\Lambda$ of the IR flow, and a UV regularisation of the theory as used in (A.17) with the UV cutoff $\Lambda_{\mathrm{UV}}$ of the theory. Both are assumed to be $k$-dependent with $\vec{\Lambda}(k) = (\Lambda(k), \Lambda_{\mathrm{UV}}(k))$. In the CS-part the regulator mass $k^2$ is removed at $p^2 = \Lambda^2 < \Lambda_{\mathrm{UV}}^2$, which renders the flow finite. Moreover, the full theory is finite as quantum fluctuations are fully suppressed for $p^2 \geq \Lambda_{\mathrm{UV}}^2$. The $k$ and $\vec{\Lambda}$ flows are a combination of the flows discussed above in appendix A.2 and appendix A.3.

The $t, t_\Lambda$-flows that originates from the CS-type part in (A.29) is given by (A.27). There is no $t$-flow related to the UV-regulator, and its $t_\Lambda$-flow is the $t_\Lambda$-flow of (A.21) with $\Lambda \to \Lambda_{\mathrm{UV}}$. The scaling factors $\vec{\mathcal{D}}_k = (\mathcal{D}_k^{(\mathrm{CS})}, \mathcal{D}_k^{(\mathrm{UV})})$ can be chosen independently and we have

$$\partial_t V_k^{(1)}(\phi) = \frac{k^2}{(4\pi)^2} V_k^{(3)}(\phi)$$
$$\times \left[ 1 + \log\left(\frac{V_k^{(2)}(\phi) + k^2}{\Lambda^2}\right) - \frac{\mathcal{D}_k^{(\mathrm{CS})}}{\left(1 + \frac{k^2 + V_k^{(2)}(\phi)}{\Lambda^2}\right)\left(1 + \frac{V_k^{(2)}(\phi)}{\Lambda^2}\right)} + \frac{\overline{\mathcal{D}}_k^{(\mathrm{UV})}}{1 + \frac{V_k^{(2)}(\phi)}{\Lambda_{\mathrm{UV}}^2}} \right], \tag{A.30}$$

with $\overline{\mathcal{D}}_k^{(\mathrm{UV})} = (\Lambda_{\mathrm{UV}}^2/k^2)\mathcal{D}_k^{(\mathrm{UV})}$. Now we expand (A.30) in (inverse) powers of $\Lambda^2$ and $\Lambda_{\mathrm{UV}}^2$,

leading to

$$\partial_t V_k^{(1)}(\phi) = \frac{k^2}{(4\pi)^2} V_k^{(3)}(\phi) \left[ 1 + \log\left( \frac{V_k^{(2)}(\phi) + k^2}{\Lambda^2} \right) \right.$$

$$\left. - \mathcal{D}_k^{(\text{CS})}\left( 1 - \frac{k^2}{\Lambda^2} - 2\frac{V_k^{(2)}(\phi)}{\Lambda^2} \right) + \overline{\mathcal{D}}_k^{(\text{UV})}\left( 1 - \frac{V_k^{(2)}(\phi)}{\Lambda_{\text{UV}}^2} \right) \right]. \tag{A.31}$$

Equation (A.31) fixes the functional form of the flow of the counter term action uniquely: it is given by the sum of the $\vec{\mathcal{D}}_k^{(\text{CS})}$-terms in (A.31). Hence we are led to

$$\partial_t S_{\text{ct}}^{(1)}[\phi] = \frac{k^2}{(4\pi)^2} V_k^{(3)}(\phi) \left[ \log\frac{\Lambda_{m_\phi}^2}{\Lambda^2} + c_{\lambda_\phi} \frac{V_k^{(2)}(\phi)}{\Lambda_{m_\phi}^2} \right], \tag{A.32}$$

where $\Lambda_{m_\phi}$ and $c_{\lambda_\phi}$ are the two renormalisation parameters. Note that the both, the prefactor $V^{(3)}(\phi)$ and the term $V^{(2)}(\phi)$ are fixed by the general structure of the flow itself and not by the specific regulators used here. There are higher order terms $(V^{(2)}(\phi)\Lambda^2)^n$, whose prefactors depend on the chosen regulator. They are also present here and they only survive the $\vec{\Lambda} \to \infty$ limit for sufficiently divergent $\vec{\mathcal{D}}_k^{(\text{CS})}$. However, such choices simply entail the inclusion of higher order relevant couplings such as $\lambda_{n,k}$.

In a $\phi^4$ approximation they reduce to the standard $\partial_t \delta\lambda_\phi$ and $\partial_t \delta m_\phi^2$ parameters in a perturbative subtraction scheme with the counter term action flow $\partial_t S_{\text{ct}}^{(1)}[\phi]$ derived from (A.6). In the present local potential approximation this necessarily generalises to derivatives of the full effective potential.

The renormalisation parameters are combinations of the $\mathcal{D}_k$'s with

$$\left( 1 - \frac{k^2}{\Lambda^2} \right) \mathcal{D}_k^{(\text{CS})} - \overline{\mathcal{D}}_k^{(\text{UV})} = \log\frac{\Lambda_{m_\phi}^2}{\Lambda^2},$$

$$\frac{\Lambda_{m_\phi}^2}{\Lambda_{\text{UV}}^2}\overline{\mathcal{D}}_k^{(\text{UV})} - 2\frac{\Lambda_{m_\phi}^2}{\Lambda^2}\mathcal{D}_k^{(\text{CS})} = c_{\lambda_\phi}, \tag{A.33}$$

with the solutions

$$\mathcal{D}_k^{(\text{CS})} = -\frac{\Lambda^2}{\Lambda_{m_\phi}^2} \frac{c_{\lambda_\phi} - \frac{\Lambda_{m_\phi}^2}{\Lambda_{\text{UV}}^2}\log\frac{\Lambda^2}{\Lambda_{m_\phi}^2}}{2 - \frac{\Lambda^2 - k^2}{\Lambda_{\text{UV}}^2}},$$

$$\overline{\mathcal{D}}_k^{(\text{UV})} = -\frac{\Lambda^2}{\Lambda_{m_\phi}^2} \frac{c_{\lambda_\phi}\left(1 - \frac{k^2}{\Lambda^2}\right) + 2\frac{\Lambda_{m_\phi}^2}{\Lambda^2}\log\frac{\Lambda^2}{\Lambda_{m_\phi}^2}}{2 - \frac{\Lambda^2 - k^2}{\Lambda_{\text{UV}}^2}}. \tag{A.34}$$

We emphasise that it is not the explicit solution (A.34) that matters, but solely its existence: the latter entails that the relations (A.33) can be used for the definition of the counter term flow (A.32). Using the later in the flow (A.31) leads us to

$$\partial_t V_k^{(1)}(\phi) = \frac{k^2}{(4\pi)^2} V_k^{(3)}(\phi) \left[ 1 + \log\left( \frac{k^2 + V_k^{(2)}(\phi)}{\Lambda_{m_\phi}^2} \right) - c_{\lambda_\phi}\frac{V_k^{(2)}(\phi)}{\Lambda_{m_\phi}^2} \right], \tag{A.35}$$

our final result for the finite CS flow in the Local Potential Approximation with full flowing renormalisation with the renormalisation constants $c_{\lambda_\phi}$ (coupling) and $\Lambda_{m_\phi}$ (mass). The latter constant enters via a logarithmically divergent counter term in the flow, which reflects the fact that divergences in the CS flow are lowered by two: quadratic divergence in standard diagrams (e.g. in a DSE) leads to logarithmic divergence in the flow. Similarly, the logarithmic divergence of the coupling is reduced to a finite subtraction. Finally, for $c_{\lambda_\phi} \equiv 0$ the full flow (A.35) reduces to (A.24), and the renormalisation condition for the coupling is left free. Note that this is the standard approach to renormalisation in the flow equation: the renormalisation is implicit in the finite initial effective action which also implicitly determines the renormalisation conditions.

As already discussed above, the explicit solution of the $\vec{\mathcal{D}}_k$ has dropped out and has to drop out of the explicit solution as it concerns details of the UV regularisation that are removed in the renormalisation. However, in order to understand the respective UV flow, it is instructive to consider the asymptotic limits of $\Lambda, \Lambda_{\mathrm{UV}}$, for which the solution simplifies. In general we have

$$k^2, \Lambda_{m_\phi}^2 < \Lambda^2 \leq \Lambda_{\mathrm{UV}}^2, \tag{A.36}$$

as $k$ is the infrared cutoff scale and $\Lambda_{m_\phi}$ is a renormalisation scale. In turn, $\Lambda_{\mathrm{UV}}$ is the UV cutoff scale and hence the maximal scale in the theory, while $\Lambda$ limits the UV range of the infrared regulator. It should be larger than the IR scales, but it has been assumed to be smaller than $\Lambda_{\mathrm{UV}}$ in the derivation.

While it is only the existence of the explicit solution (A.34) that matters, it is still instructive to evaluate its properties. To that end we discuss them within asymptotic choice of the UV cutoffs. A natural choice is $\Lambda \to \Lambda_{\mathrm{UV}}$ and consequently $\Lambda \to \infty$. This procedure implements the CS limit for all $\Lambda$, the IR mass $k^2$ is present for all momentum scales in the theory with $p^2 \leq \Lambda_{\mathrm{UV}}$. In this limit (A.34) reduces to

$$\mathcal{D}_k^{(\mathrm{CS})} = -c_{\lambda_\phi} \frac{\Lambda^2}{\Lambda_{m_\phi}^2} + \log \frac{\Lambda^2}{\Lambda_{m_\phi}^2},$$

$$\overline{\mathcal{D}}_k^{(\mathrm{UV})} = -c_{\lambda_\phi} \frac{\Lambda^2}{\Lambda_{m_\phi}^2} + 2\log \frac{\Lambda^2}{\Lambda_{m_\phi}^2}. \tag{A.37}$$

The scalings of the two terms with $c_{\lambda_\phi}$ and $\Lambda_{m_\phi}$ (inversely) reflect the UV relevance of the respective couplings: the renormalisation constant $c_{\lambda_\phi}$ is linked to marginal coupling $\lambda_{\phi,k}$. As the infrared flow lowers the UV degree of divergence by two, this is reinstated by the multiplication with $\Lambda^2$. Note also, that $\mathcal{D}_k^{(\mathrm{UV})} = k^2/\Lambda^2 \overline{\mathcal{D}}_k^{(\mathrm{UV})}$ and hence has a finite limit. This originates in the fact, that the respective $\Lambda$-flow is a UV flow and its UV degree of divergence is not lowered.

We note in passing, that a theory with fundamental $\lambda_n$ couplings would require $\mathcal{D}_k$'s, that diverge with $\Lambda^{n-2}$. While such flows can be defined, there UV consistency is at stake. We emphasise that this has but nothing to do with the CS setting, but rather with the potential non-renormalisability of these theories. Already in the $\phi^4$ case discussed here we only derived consistent flows. Studying the UV closure of the theory for $k \to \infty$ in the present truncation enforces the triviality of the $\phi^4$ theory. We rush to add that this is not a triviality proof as it is obtained in a truncation.

## A.5 Flowing renormalisation at work

We close this discussion with a simple example of the *flowing renormalisation*, or rather a non-flowing one. For the sake of simplicity we restrict ourselves to theories in the symmetric phase

with $\phi_{\text{EoM}} = 0$. Furthermore we chose the renormalisation point $\mu = 0$. Then the flow of the mass $m^2_{\phi,k} = V^{(2)}(0)$ and the coupling $\lambda_{\phi,k} = V^{(4)}(0)$ is given by

$$\dot{m}^2_{\phi,k} = \frac{k^2}{(4\pi)^2} \lambda_{\phi,k} \left[ 1 - c_{\lambda_\phi} \frac{m^2_{\phi,k}}{\Lambda^2_{m_\phi}} + \log\left( \frac{k^2 + m^2_{\phi,k}}{\Lambda^2_{m_\phi}} \right) \right],$$

$$\dot{\lambda}_{\phi,k} = \frac{1}{(4\pi)^2} \left\{ \lambda_{6,k} k^2 \left[ 1 - c_{\lambda_\phi} \frac{m^2_{\phi,k}}{\Lambda^2_{m_\phi}} + \log\left( \frac{k^2 + m^2_{\phi,k}}{\Lambda^2_{m_\phi}} \right) \right] + 3\lambda^2_{\phi,k} \left( \frac{1}{1 + \frac{m^2_{\phi,k}}{k^2}} - c_{\lambda_\phi} \frac{k^2}{\Lambda^2_{m_\phi}} \right) \right\},$$

where $\lambda_{6,k} = V^{(6)}(0)$ with (A.5). We have also used that $V^{(2n+1)}(0) = 0$ for all $n \in \mathbb{N}$ as the effective action is invariant under $\phi \to -\phi$. In terms of diagrams the first term in the flows $\dot{m}^2_{\phi,k}$ and $\dot{\lambda}_{\phi,k}$ is the contribution of the respective tadpole diagram being proportional to $\lambda_{n+2,k}$ for the flow of $\lambda_{n,k}$. At $\phi = 0$ this is the only diagram that contributes to the flow of the mass and for $\dot{m}^2_{\phi,k} = 0$ we have to choose

$$\Lambda^2_{\text{ref}} = \left( m^2_{\phi,k} + k^2 \right) \exp\left\{ 1 - c_{\lambda_\phi} \frac{m^2_{\phi,k}}{\Lambda^2_{\text{ref}}} \right\}, \tag{A.38}$$

which also eliminates the tadpole contributions in the flow of all $\lambda_{n,k}$. In turn, the flow of the coupling also contains the fish diagram proportional to $\lambda^2_{\phi,k}$. In perturbation theory this diagram is logarithmically divergent while it is finite for the CS flow. The (re)-normalisation of this diagram is linked to $c_{\lambda_\phi}$ and $\dot{\lambda}_{\phi,k} = 0$ is achieved by setting the expression in parenthesis in the flow of $\lambda_{\phi,k}$ to zero. This leads us to

$$c_{\lambda_\phi} = \frac{\Lambda^2_{m_\phi}}{k^2} \frac{1}{1 + \frac{m^2_{\phi,k}}{k^2}}, \tag{A.39}$$

and inserting (A.39) in (A.38) leads to

$$c_{\lambda_\phi} = \exp\left\{ \frac{1}{1 + \frac{m^2_\phi}{k^2}} \right\},$$

$$\Lambda^2_{\text{ref}} = \left( m^2_\phi + k^2 \right) \exp\left\{ \frac{1}{1 + \frac{m^2_\phi}{k^2}} \right\}, \tag{A.40}$$

where we also used that $m_{\phi,k} = m_\phi$ for all $k$. The two renormalisation constants have the limits

$$c_{\lambda_\phi}(k \to 0) = 1, \qquad \Lambda^2_{\text{ref}}(k \to 0) = m^2_\phi. \tag{A.41}$$

Interestingly, the two renormalisation parameters know nothing about the couplings of the theory owing to the peculiarity that the mass renormalisation only involved that tadpole. While this property is approximation-independent, the factorisation of the couplings from the loop only holds in approximations with momentum-independent couplings.

# B Decomposition of the fermion propagator for spatial regulators

Spatial regulators break Lorentz symmetry by construction. In this case, the fermionic two-point function may be parametrised as

$$\Gamma^{(2)}_{\psi\bar\psi} = Z_\psi(p)\Big[i\gamma_0 p_0 + M_\psi(p)\Big] + Z^\perp_\psi(p)\, i\vec\gamma\cdot\vec p\,, \tag{B.1}$$

where we have used the notation $Z_\psi(p) \equiv Z^\parallel_\psi(p)$ for the longitudinal dressing and $Z^\perp_\psi(p)$ for the transverse one, respectively. Our choice guarantees that $M_\psi(p)$ is related to the pole mass. Through (2) this again leads to a fermionic propagator which can be cast into the form of (84). With the regulators defined in (79) for $\epsilon = \epsilon(\vec p)$ and (81), the Dirac, scalar and universal part of the propagator read

$$G^{(d)}_\psi(p) = \frac{1}{\not p}\left[\gamma_0 p_0 + \vec\gamma\cdot\vec p\left(A_\psi(p) + \frac{A^0_\psi(p)}{\sqrt{\vec p^2}}\bar R^\psi_d\right)\right]G^{(u)}_\psi\,, \tag{B.2}$$

$$G^{(s)}_\psi(p) = \frac{1}{M_\psi}\Big[M_\psi(p) + A^0_\psi(p)\bar R^\psi_s\Big]G^{(u)}_\psi\,, \tag{B.3}$$

and

$$G^{(u)}_\psi(p) = \frac{1}{Z_\psi(p)}\frac{1}{p_0^2 + \left(\sqrt{\vec p^2}A_\psi(p) + A^0_\psi(p)\bar R^\psi_d\right)^2 + \left(M_\psi(p) + A^0_\psi(p)\bar R^\psi_s\right)^2}\,. \tag{B.4}$$

Here, we have made use of the following definitions,

$$A_\psi(p) = \frac{Z^\perp_\psi(p)}{Z_\psi(p)}\,, \qquad A^0_\psi(p) = \frac{Z_\psi}{Z_\psi(p)}\,. \tag{B.5}$$

Note that $A^0_\psi(p)$ depends on the choice for the momentum-independent prefactor of the regulators. In this appendix, we have used $Z_\psi \equiv Z_\psi(0)$ for all tensor structures.

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
