# Peer review of "Renormalised spectral flows"

_SciPost Physics, doi:SciPost Phys. Core 6, 061 (2023)_

## Round 2 · Referee Report · Anonymous (Referee 1) · 2023-1-17

Report
(i) Essential arguments, e.g. (17), (19) and (27), in Section IIIA rely on Ref. [30]. The authors should specify where the derivations of those equations are discussed in Ref. [30], like section ?? in [30] and equation ?? in [30].
(ii) Very minor things:
The last paragraph Section IIID, “one may has to” -> “one may have to”.
The abbreviation of the spacetime integral already is defined below Eq.(17), so that in Eqs.(63) and (A1) can be abbreviated.
What is the measure of integral in Eq.(A4)?

---

## Round 3 · Author Response

Incorporates the suggestions from the referee report

---

## Round 3 · List of Changes

• Added several remarks regarding the relation to [32] (formerly [30]), including more detailed chapter references
  • Fixed typos

---

## Editorial Decision

published